# SlaClip: Gradient Norm Slacks can be Indicator for Adaptive Clipping in DP-SGD

Shuyan Zou [1]   Shaowei Wang [2]   Zhanxing Zhu [1]   Jin Li [2]   Changyu Dong [2]   Vladimiro Sassone [1]   Han Wu [*1]

## Abstract

Differentially private stochastic gradient descent (DP-SGD) achieves privacy by clipping per-sample gradients and injecting Gaussian noise, but its utility is highly sensitive to the choice of the clipping threshold $C$. A fixed $C$ often degrades performance and necessitates repeated empirical calibration. Existing adaptive clipping methods either modify the gradient update in vanilla DP-SGD, causing additional tuning or optimization overhead, or introduce separate private queries to monitor gradient statistics. In contrast, we leverage the *slack* information induced by the standard clipping operation, an overlooked signal in prior work, and show that it provides an effective indication for adapting $C$. In light of this, we propose *SlaClip*, a privacy-preserving adaptive clipping strategy using a post-hoc *Slack Indicator*. Under the same training configuration and privacy accountant, *SlaClip* preserves the sampling rule, noise multiplier, and global $\ell_2$ sensitivity bound of vanilla DP-SGD. Therefore, *SlaClip* is a plug-and-play module for vanilla DP-SGD and its variants. Moreover, *SlaClip* is accounted under the same per-step privacy bound, while requiring no additional private query. Across diverse datasets and tasks, experiments show that *SlaClip* consistently outperforms baseline adaptive clipping methods.

## 1. Introduction

Differentially private stochastic gradient descent (DP-SGD) (Abadi et al., 2016) is a standard approach for training

*Project lead. [1]Emails: {s.zou,z.zhu,vsassone}@soton.ac.uk, University of Southampton, Southampton, United Kingdom. [2]Emails: {lijin,changyu.dong}@gzhu.edu.cn, Guangzhou University, Guangzhou, Guangdong, China. Corresponding authors: Shaowei Wang <wangsw@gzhu.edu.cn>, Han Wu <h.wu@soton.ac.uk>.

*Proceedings of the $43^{rd}$ International Conference on Machine Learning*, Seoul, South Korea. PMLR 306, 2026. Copyright 2026 by the author(s).

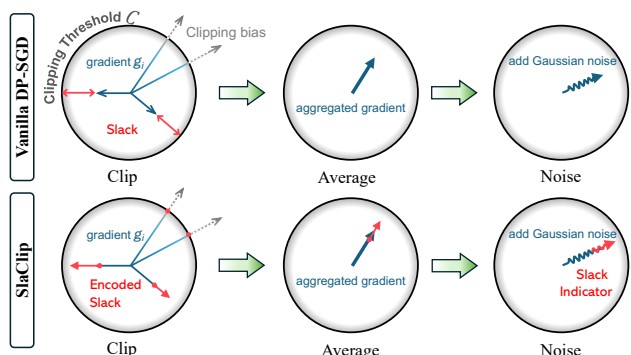

*Figure 1.* Overview of *SlaClip* within the vanilla DP-SGD pipeline. Both vanilla DP-SGD and *SlaClip* follow the same clip-average-noise pipeline. *SlaClip* extends gradients by encoding slack information during the clipping step. The extended gradients preserve the original $\ell_2$ norm sensitivity bound, enabling the Slack Indicator to be released through the same Gaussian mechanism without an additional private query.

deep models under differential privacy (DP). At each iteration, DP-SGD samples a minibatch of training examples and computes per-sample gradients, which inherently encode private information from individual data points. Particularly, DP-SGD clips each gradient to a threshold $C$, aggregates the clipped gradients over the minibatch (typically by averaging), and perturbs the aggregate with Gaussian noise calibrated to the resulting sensitivity bound, as illustrated by the vanilla DP-SGD branch in Fig. 1. This noisy aggregate is then released as a *differentially private update*. The model then updates its parameters using this noisy aggregate, and privacy loss composes over iterations.

Vanilla DP-SGD employs a fixed clipping threshold, typically selected via empirical calibration. However, gradient norm distributions are non-stationary and evolve over training, so a fixed threshold can become misaligned with the distribution: when substantial gradient norms lie above the threshold, informative gradients are heavily truncated, when few gradient norms exceeds it, the injected noise dominates the update. This motivates adaptive clipping mechanisms that track those dynamics during training.

A natural approach is to make the clipping threshold iteration-dependent, denoted by $C_t$ at iteration $t$. Existing

methods achieve this in two ways. One line of work adapts $C_t$ by privately estimating gradient norm statistics (e.g., quantiles or distributional summaries) via additional private queries (Andrew et al., 2021; Wei et al., 2025). Another avoids such estimation by introducing other optimization components (e.g., normalization rules or clipping schedules), thereby relying on additional hyperparameter tuning, or requiring pretraining (Pichapati et al., 2019; Bu et al., 2023; Gilani et al., 2025). Both lines introduce overhead beyond vanilla DP-SGD, leaving the following open question:

*Can adaptive clipping be achieved within the vanilla DP-SGD release, without introducing additional private queries or gradient transformations?*

This paper answers this question affirmatively by proposing *SlaClip*, which obtains a noisy statistical summary of gradient norms below the current threshold $C_t$ without introducing any private query beyond the main DP-SGD release, and updates $C_t$ accordingly.

Fig. 1 illustrates the intuition behind *SlaClip*: vanilla DP-SGD leaves the clipping induced *slack* information unused, whereas *SlaClip* encodes this slack into extra coordinates of the gradient vector and releases the resulting extended gradient through the same Gaussian mechanism used for the DP-SGD update. The encoding preserves the original global $\ell_2$ sensitivity bound (proof in Lemma 3.2), so the released slack coordinates provide a noisy *Slack Indicator*: a privacy-preserving, binned estimate of the cumulative distribution function (CDF) of gradient norms on $[0, C_t]$. This CDF estimate provides both near threshold and signal from small gradients for adapting $C_t$, without introducing an additional private query. Moreover, *SlaClip* derives the Slack Indicator from the additional coordinates of the same Gaussian release, while leaving the gradient update coordinates unchanged. This makes *SlaClip* a plug-and-play module for vanilla DP-SGD and for its variants that do not already implement adaptive clipping. Our contributions are threefold:

- We propose the *Slack Indicator*, a privacy-preserving signal from the main DP-SGD Gaussian release that provides a noisy, discretized CDF estimate of gradient norms below $C_t$.

- We develop *SlaClip*, an adaptive clipping strategy for DP-SGD driven by the *Slack Indicator*.

- We empirically show that *SlaClip* is competitive with, and often improves utility over, existing methods under matched privacy budgets.

## 2. Revisiting DP-SGD

**Differential Privacy (DP)** ensures that the output of a dataset analysis query (e.g., average) is nearly equally likely whether any single individual's sample is included or not (Dwork & Roth, 2014). Formally, a mechanism $\mathcal{M}$ is $(\varepsilon, \delta)$-differentially private if for any adjacent datasets $D \sim D'$ and any measurable set $S$,

$$\Pr[\mathcal{M}(D) \in S] \ \leq \ e^\varepsilon \Pr[\mathcal{M}(D') \in S] + \delta. \quad (1)$$

where $\varepsilon$ controls the distinguishability between outputs on adjacent datasets and $\delta$ is a small failure probability. This guarantee can be achieved by adding calibrated Gaussian noise to the query output before its release. In particular, if the query produces a $d$-dimensional output $f(D)$ (e.g., an averaged gradient vector), the *Gaussian mechanism* releases

$$\mathcal{M}(D) \triangleq f(D) + \mathcal{N}\big(\mathbf{0}, \ (\sigma \, \Delta_2(f))^2 \mathbf{I}_d\big), \quad (2)$$

where $\sigma$ is a data-independent noise multiplier and $\mathbf{I}_d$ denotes the $d$-dimensional identity matrix. The global $\ell_2$ sensitivity $\Delta_2(f)$ measures the maximum influence that a single record can have on $f(D)$:

$$\Delta_2(f) \triangleq \sup_{D \sim D'} \big\| f(D) - f(D') \big\|. \quad (3)$$

The Gaussian mechanism is calibrated to this global $\ell_2$ sensitivity, which measures the largest possible change in the vector-valued query under one record perturbation.

**DP-SGD** (Abadi et al., 2016) applies DP to high-dimensional, gradient-based queries derived from individual training samples, and therefore follows the same $\ell_2$ norm sensitivity framework. It includes a key extension:

**Sensitivity control via clipping**. Unlike typical DP queries that assume a bounded sensitivity in dataset, gradients produced by training samples can vary substantially in magnitude. As a result, the $\ell_2$ norm sensitivity can become very large, requiring significant noise (Eq. (2)), which may eventually dominate the useful gradient updates and impede model convergence. DP-SGD addresses this issue via *per-sample $\ell_2$ clipping*, which bounds individual gradient contributions and controls the sensitivity $\Delta_2(f)$.

Formally, DP-SGD can be described in four steps. The first three steps are illustrated in Figure 1; the fourth step tracks the accumulated privacy loss and applies the model update as post-processing.

**Step I: Gradient computation.** This step is identical to the gradient computation performed in standard SGD: at iteration $t$, a minibatch $\mathcal{B}_t$ is sampled according to the specified sampling rule. For exposition, we write $q$ for the sampling rate used by the privacy accountant and $B = q \cdot |D|$ for the nominal, or expected, batch size. Under Poisson subsampling, the realized minibatch size $|\mathcal{B}_t|$ may vary across iterations, while $B$ is used as the fixed normalization constant in the Gaussian release. The sampled examples produce per-sample gradients $\mathbf{g}_{t,i} \in \mathbb{R}^d$ for $i \in \mathcal{B}_t$.

**Step II: Per-sample $\ell_2$ clipping.** Given a clipping threshold $C_t > 0$ (vanilla DP-SGD uses a fixed $C_t \equiv C_0 > 0$), it applies per-sample $\ell_2$ clipping for all $i \in \mathcal{B}_t$:

$$Clip_{C_t}(\mathbf{g}_{t,i}) = \begin{cases} \mathbf{g}_{t,i}, & \|\mathbf{g}_{t,i}\| \le C_t, \\ C_t \cdot \mathbf{g}_{t,i}/\|\mathbf{g}_{t,i}\|, & \|\mathbf{g}_{t,i}\| > C_t. \end{cases} \quad (4)$$

**Step III: Aggregate, noise and release.** Since Step II enforces $\|Clip_{C_t}(\mathbf{g}_{t,i})\| \le C_t$ for all $i \in \mathcal{B}_t$, the per-iteration average query on gradients

$$f_{avg}(D) \triangleq \frac{1}{B} \sum_{i \in \mathcal{B}_t} Clip_{C_t}(\mathbf{g}_{t,i})$$

has bounded global $\ell_2$ sensitivity under the add/remove neighboring relation:

$$\Delta_2(f_{avg}) = \sup_{D \sim D'} \left\| f_{avg}(D) - f_{avg}(D') \right\| \le C_t/B.$$

Therefore, the clipped gradients are averaged and perturbed with Gaussian noise calibrated to this sensitivity:

$$\widetilde{\mathbf{g}}_t = \frac{1}{B} \sum_{i \in \mathcal{B}_t} Clip_{C_t}(\mathbf{g}_{t,i}) + \mathcal{N}\left(\mathbf{0}, \, (\frac{\sigma C_t}{B})^2 \mathbf{I}_d\right). \quad (5)$$

Upon completion of this step, $\widetilde{\mathbf{g}}_t$ constitutes a *differentially private gradient release*.

**Step IV: Privacy accounting and model update.** The release in Step III incurs privacy loss, which is tracked by a privacy accountant. In our exposition and experiments, we use the common Rényi differential privacy (RDP) accountant (Mironov, 2017) as an instantiation. Given a specified sampling scheme (e.g., Poisson subsampling) and hyperparameters, namely the sampling rate $q$, noise multiplier $\sigma$, and Rényi order $\alpha > 1$, the privacy accountant computes the per-step RDP parameter by bounding the Rényi divergence between the output distributions on adjacent datasets. Specifically, for two probability distributions $P$ and $Q$, the order-$\alpha$ Rényi divergence is

$$D_\alpha(P \| Q) \triangleq \frac{1}{\alpha - 1} \log \int \left(\frac{dP}{dQ}\right)^\alpha dQ.$$

A mechanism satisfies $(\alpha, \varepsilon_\alpha)$-RDP if, for all adjacent datasets, the order-$\alpha$ Rényi divergence between the corresponding output distributions is at most $\varepsilon_\alpha$. In DP-SGD, the accountant bounds this order-$\alpha$ divergence for the subsampled Gaussian mechanism at each iteration, yielding the corresponding per-step RDP guarantee.

Updating model parameters $\boldsymbol{\theta}_{t+1} = \boldsymbol{\theta}_t - \text{lr} \cdot \widetilde{\mathbf{g}}_t$ is post-processing of a differentially private release and therefore does not affect the privacy guarantee (Dwork & Roth, 2014). DP-SGD then proceeds to iteration $t+1$ and repeats Steps I–IV while tracking the cumulative privacy loss until the pre-specified privacy budget is exhausted.

## 3. SlaClip

**Motivation.** This paper considers DP-SGD under a fixed configuration setting, denoted Reg*, in which all training and privacy configurations are specified a priori, including the dataset domain $\mathcal{D}$, the neighboring relation $\sim$, the sampling rule, the nominal batch size or sampling rate, the noise multiplier $\sigma > 0$, and the privacy accounting rule. Within such a fixed configuration, the clipping threshold remains the primary degree of freedom affecting model utility. Vanilla DP-SGD uses a fixed clipping threshold $C_t \equiv C_0$, chosen typically via empirical calibration. However, the gradient norm distributions are non-stationary and evolve over iterations, so a fixed $C_t$ cannot remain well aligned with the training dynamics. As shown in Eq. (4), when a substantial fraction of $\|\mathbf{g}_{t,i}\|$ lies above $C_t$, many informative gradients are truncated, causing excessive clipping; when few samples' $\|\mathbf{g}_{t,i}\|$ exceed $C_t$, the noise injected in Eq. (5) can dominate the gradient update. This motivates *adaptive clipping* approaches that respond to distributional dynamics during training.

Our approach follows the intuition of Google's Adap-Clip (Andrew et al., 2021): estimating the fraction of clipped samples at iteration $t$ provides a feedback signal for updating the clipping threshold $C_{t+1}$, allowing the threshold to evolve adaptively during training.

However, Adap-Clip introduces *an additional* per-iteration private query (bit sum) to count the clipped samples. Under a fixed privacy accountant, this extra query requires additional privacy accounting, and maintaining the same target privacy budget typically necessitates either stronger noise or privacy budget reallocation across releases. Subsequent work (Wei et al., 2025) follows the same additional private query design pattern by re-balancing noise across multiple releases under a fixed privacy budget.

We note that this line of work underexplored an inherent property of the Gaussian releases in high dimension, formalized in Theorem 3.1 below. We show that this property can be leveraged to estimate the gradient norm distribution without introducing any private query beyond the main DP-SGD release. This insight leads to *SlaClip*, a single-release adaptive clipping method.

The following result formalizes the sensitivity-preserving extension principle used by *SlaClip*. The principle is not tied to RDP (Mironov, 2017): if the extended query preserves the original global $\ell_2$ sensitivity bound and uses the same Gaussian noise multiplier under the same sampling rule, then it is passed to a subsampled Gaussian accountant with the same per-step accounting parameters. For concreteness, we instantiate the statement with the RDP bound for the Gaussian mechanism.

**Theorem 3.1** (Extension of the Gaussian Mecha-

nism (Dwork & Roth, 2014)). *Extending a Gaussian query with additional, possibly informative coordinates does not change the Gaussian mechanism RDP upper bound when the extension preserves the original query's global $\ell_2$ sensitivity.*

*Let $f : \mathcal{D} \to \mathbb{R}^d$, $f^+ : \mathcal{D} \to \mathbb{R}^{d+K}$ be deterministic query functions on $\mathcal{D}$, where $d$ and $d + K$ denote the output dimensions. $D \sim D'$ are adjacent datasets. Let $\mathrm{D}_\alpha(*\|*)$ denote the Rényi divergence with order $\alpha$. If $\Delta_2(f) = \Delta_2(f^+) = \Delta$, then the Gaussian mechanism satisfies the same RDP guarantee for $f$ and $f^+$:*

$$\sup_{D \sim D'} \mathrm{D}_\alpha\big(\mathcal{N}(f(D), (\sigma\Delta)^2 \mathbf{I}_d)\|\mathcal{N}(f(D'), (\sigma\Delta)^2 \mathbf{I}_d)\big),$$
$$\sup_{D \sim D'} \mathrm{D}_\alpha\big(\mathcal{N}(f^+(D),(\sigma\Delta)^2\mathbf{I}_{d+K})\|\mathcal{N}(f^+(D'),(\sigma\Delta)^2\mathbf{I}_{d+K})\big).$$

*Proof.* Based on the exact Rényi divergence bound (Mironov, 2017) on Gaussian noise, we have the divergence $\mathrm{D}_\alpha(\mathcal{N}(f(D), (\sigma\Delta)^2\mathbf{I}_d\|\mathcal{N}(f(D'), (\sigma\Delta)^2\mathbf{I}_d)$ equals to $\alpha \cdot \|f(D) - f(D')\|^2/(2\sigma^2\Delta^2)$, and the $\mathrm{D}_\alpha(\mathcal{N}(f^+(D), (\sigma\Delta)^2\mathbf{I}_{d+K})\|\mathcal{N}(f^+(D'),(\sigma\Delta)^2\mathbf{I}_{d+K}))$ equals to $\alpha \cdot \|f^+(D) - f^+(D')\|^2/(2\sigma^2\Delta^2)$. By assumption, we have $\Delta_2(f) = \sup_{D \sim D'} \|f(D) - f(D')\|$ equals to $\Delta_2(f^+) = \sup_{D \sim D'} \|f^+(D) - f^+(D')\|$, then we obtain the conclusion. □

**Overview.** We design *SlaClip* by exploiting Theorem 3.1 as a *design principle*: within a single DP-SGD iteration, one may release a higher-dimensional vector in place of the vanilla noised average gradient while preserving the same Gaussian-mechanism privacy bound, provided the resulting query preserves global $\ell_2$ sensitivity, i.e., $\Delta_2(f_{avg}) = \Delta_2(f_{avg}^+)$. Here, $f_{avg}$ and $f_{avg}^+$ denote the average queries over the vanilla and extended gradients, respectively, within a single DP-SGD iteration and with a single Gaussian release.

A simple instantiation of this principle would mirror Adap-Clip: append a per-sample clipped/unclipped binary indicator as an extra dimension and let DP-SGD aggregate and noise it in the same DP-SGD Gaussian release. However, we note Theorem 3.1 is more permissive: the Gaussian-mechanism privacy cost bound is dimension independent and does not depend on $K$, provided the extension preserves the original global $\ell_2$ sensitivity used for calibration. Motivated by this, we revisit the DP-SGD pipeline to identify a signal that can be encoded into extra coordinates while (i) introducing no private query beyond the main DP-SGD release, (ii) maintaining $\Delta_2(f_{avg}) = \Delta_2(f_{avg}^+)$, and (iii) providing informative summaries of $\|\mathbf{g}_{t,i}\|$ distribution.

We show that such a signal exists and is naturally available within DP-SGD, which we term the *slack*, i.e., the below threshold gap $(C_t - \|\mathbf{g}_{t,i}\|)_+$ between the current threshold and the per-sample gradient norm. We now describe how *SlaClip* implements the above principles.

**Step 1: Gradient computation.** This step is identical to vanilla DP-SGD step I (Section 2): compute per-sample gradients $\mathbf{g}_{t,i} \in \mathbb{R}^d$ for $i \in \mathcal{B}_t$. We note another line of adaptive clipping methods transforming the gradients at this step, often introducing additional hyperparameters or require pretraining (Bu et al., 2023; Gilani et al., 2025). *SlaClip* follows a different design direction and is evaluated against representative methods from both lines in Section 4.

**Step 2: Clipping and Slack encoding.** In the per-sample gradient clipping process of Eq. (4), clipping also induces *slack information*, namely the unused norm margin between the clipping threshold and the gradient norm, given by $\max\{ C_t - \|\mathbf{g}_{t,i}\|, 0 \}$. Fig. 2-B provides a simple illustration of this slack information.

*SlaClip* encodes this slack information without interfering with the vanilla DP-SGD procedure. For each per-sample gradient $\mathbf{g}_{t,i} \in \mathbb{R}^d$, *SlaClip* appends a $K$-dimensional vector to the original gradient:

$$\mathbf{g}_{t,i}^+ = \big[\, Clip_{C_t}(\mathbf{g}_{t,i});\ \mathbf{s}_{t,i} \,\big]. \tag{6}$$

Here, $\mathbf{g}_{t,i}^+$ is the extended $(d + K)$-dimensional gradient, and $\mathbf{s}_{t,i}$ is a *slack vector* that encodes the above slack information and is defined as

$$\mathbf{s}_{t,i} \triangleq \big[\lambda \mathbf{1}^{(a)};\ b;\ \mathbf{0}\big]_{t,i}, \tag{7}$$

where $\lambda \triangleq C_t/\sqrt{K}$ and $\mathbf{1}^{(a)}$ denotes an $a$-dimensional all-ones vector. The parameters $a \in \mathbb{N}$ and $b \in [0, \lambda)$ are uniquely determined by

$$\sqrt{K} \cdot \max\{ C_t - \|\mathbf{g}_{t,i}\|, 0 \} = a\lambda + b. \tag{8}$$

The choice of $\lambda$ is dictated by norm geometry: since each coordinate of $\mathbf{s}_{t,i}$ has magnitude at most $\lambda$, we have $\|\mathbf{s}_{t,i}\|_2 \leq \sqrt{K} \lambda = C_t$. This coordinate wise cap enables fine-grained encoding while keeping the slack component uniformly bounded; together with the construction in Eq. (6), it yields the full per-sample bound $\|\mathbf{g}_{t,i}^+\| \leq C_t$ proved in Lemma 3.2.

**Lemma 3.2** (Per-sample $\ell_2$ bound of extended gradient). *For all $i \in \mathcal{B}_t$, the extended gradient given in Eq. (6) satisfies $\|\mathbf{g}_{t,i}^+\| \leq C_t$. Consequently, under add/remove adjacency, the $\ell_2$ sensitivity of the* average *query satisfies $\Delta_2(f_{avg}^+) = C_t/B = \Delta_2(f_{avg})$. (Full proof in Appendix Lemma. B.2)*

*Proof Sketch.* If $\|\mathbf{g}_{t,i}\| > C_t$, Eq. (6) gives

$$\|\mathbf{g}_{t,i}^+\| = \|[C_t \cdot \mathbf{g}_{t,i}/\|\mathbf{g}_{t,i}\|; \mathbf{0}]\| = C_t.$$

If $\|\mathbf{g}_{t,i}\| \leq C_t$, then by Eq. (7), and $\lambda = C_t/\sqrt{K} \leq C_t$,
$$\|\mathbf{g}_{t,i}^+\|^2 = \|\mathbf{g}_{t,i}\|^2 + a\lambda^2 + b^2 \leq \|\mathbf{g}_{t,i}\|^2 + \lambda(a\lambda + b)$$
$$\leq \|\mathbf{g}_{t,i}\|^2 + C_t(C_t - \|\mathbf{g}_{t,i}\|) \leq C_t^2,$$

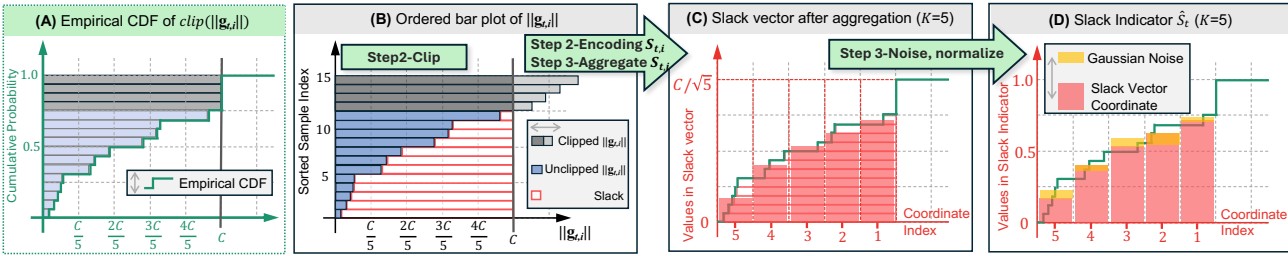

*Figure 2.* Illustration of the Slack Indicator as a binned CDF estimator. (**A**) The empirical CDF of clipped gradients' $\ell_2$ norms is the reference target: it is not directly queried, but represents the distributional information that the Slack Indicator aims to estimate. (**B–D**) SlaClip obtains this estimate through slack encoding, aggregation, and noisy release. (**B**) Ordered per-sample gradient norms with clipping at threshold $C_t$, where slack is the gap between $C_t$ and the unclipped norm. (**C**) Aggregated $K = 5$ dimensional slack vector in reversed index order, with each coordinate capped at $\lambda = C/\sqrt{5}$ and adjacent coordinates corresponding to gradient norm bins of width $C/5$. (**D**) The Slack Indicator $\hat{\mathbf{s}}_t$, obtained after Gaussian noise and normalization, provides a privacy-preserving, binned estimate of the reference CDF in (**A**).

implying $\|\mathbf{g}_{t,i}^+\| \leq C_t$. Under $\mathcal{B} \sim \mathcal{B}'$ add/remove adjacency, the average query satisfies

$$\Delta_2(f_{avg}^+) = \sup_{\mathcal{B} \sim \mathcal{B}'} \|f_{avg}^+(\mathcal{B}) - f_{avg}^+(\mathcal{B}')\| = \frac{C_t}{B} = \Delta_2(f_{avg}).$$

**Step 3: Single-release Gaussian release.** As part of the extended gradient, slack vectors inherit the privacy guarantee of the vanilla DP-SGD Gaussian mechanism:

$$\widetilde{\mathbf{g}}_t^+ = \overbrace{\frac{1}{B} \sum_{i \in \mathcal{B}_t} \mathbf{g}_{t,i}^+}^{f_{avg}^+} + \mathcal{N}\left(\mathbf{0}, \left(\frac{\sigma C_t}{B}\right)^2 \mathbf{I}_{d+K}\right). \quad (9)$$

Fig. 2-B–D provide a simple example illustrating the above steps. This uses no additional private query beyond the main DP-SGD release: the extended average query $f_{avg}^+$ and the slack summary are released together in a single Gaussian release. By Lemma 3.2, the extended averaged query $f_{avg}^+$ preserves the original global $\ell_2$ sensitivity calibration $C_t/B$ of the vanilla averaged clipped gradient query. Hence, under the same sampling rule, noise multiplier, and privacy accountant, *SlaClip* is accounted with the same per-step privacy cost upper bound as vanilla DP-SGD. Theorem 3.1 instantiates this dimension extension argument with the RDP bound for the Gaussian mechanism.

In contrast, methods such as Adap-Clip (Andrew et al., 2021) introduce additional private queries at this step, which require additional privacy accounting; under the same target privacy budget, this typically necessitates stronger noise or privacy budget reallocation across releases.

Writing the last $K$ coordinates explicitly, the differentially private release is given by

$$\widetilde{\mathbf{g}}_t^+ = [\widetilde{\mathbf{g}}_t; \widetilde{\mathbf{s}}_t], \quad (10)$$

where $\widetilde{\mathbf{g}}_t$ is identical to the noised gradient released by vanilla DP-SGD and is used for the model update, while $\widetilde{\mathbf{s}}_t$ is the released slack summary. *SlaClip* further builds the *Slack Indicator* by normalizing it as $\hat{\mathbf{s}}_t \triangleq \widetilde{\mathbf{s}}_t/\lambda$ to guide

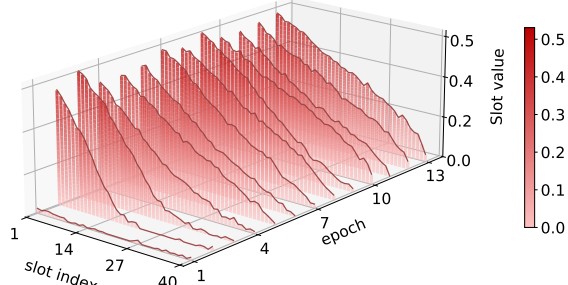

*Figure 3.* Visualization of the released *Slack Indicator* profiles over training on CIFAR-10 with $K = 40$. For visual clarity, we plot only the profile from one minibatch of each epoch. Each profile gives a noisy, binned estimate of the CDF of gradient norms on $[0, C_t]$. The first coordinate, corresponding to the bin nearest $C_t$, quickly stabilizes around $0.5$ and provides feedback for threshold adaptation, while the last coordinate reflects the increasing mass of small-norm gradients near zero.

clipping threshold adaptation.

**Statistical Interpretation of Slack Indicator $\hat{\mathbf{s}}_t$.** As illustrated in Fig. 2, the normalized release $\hat{\mathbf{s}}_t$ can be viewed as a noisy, binned estimate of the cumulative distribution function (CDF) of clipped gradient $\ell_2$ norms $\|\mathrm{clip}_{C_t}(\mathbf{g}_{t,i})\|$ over the minibatch. Each slack coordinate is capped at $\lambda = C_t/\sqrt{K}$; after mapping back to gradient norm space, adjacent coordinates correspond to equal-width bins of width $C_t/K$. Specifically, the $k$-th coordinate $\hat{s}_{t,k}$ estimates a bin-averaged CDF value over the interval $[C_t - kC_t/K, C_t - (k-1)C_t/K]$.

Formally, the normalized released coordinate can be written as

$$\hat{s}_{t,k} = \frac{K}{BC_t} \sum_{i \in \mathcal{B}_t} \int_{C_t - kC_t/K}^{C_t - (k-1)C_t/K} \mathbf{1}\{\|\mathbf{g}_{t,i}\| \leq u\} \, du + \varepsilon_{t,k},$$
$$(11)$$

where $\varepsilon_{t,k}$ denotes the normalized Gaussian noise. Equiv-

alently, $\varepsilon_{t,k} = \xi_{t,k}/\lambda$ with $\xi_{t,k} \sim \mathcal{N}(0, (\sigma C_t/B)^2)$. Ignoring the zero-mean Gaussian noise, $\mathbb{E}[\hat{s}_{t,k}]$ lies between the endpoint CDF values:

$$\left[ \Pr\left( \|\mathbf{g}_{t,i}\| \leq C_t - \frac{kC_t}{K} \right), \Pr\left( \|\mathbf{g}_{t,i}\| \leq C_t - \frac{(k-1)C_t}{K} \right) \right].$$

The above interpretation implies that each minibatch processed by *SlaClip* yields a privacy-preserving, binned CDF estimate of gradient norms on $[0, C_t]$, providing richer information than a single clipped/unclipped statistic. Fig. 3 visualizes such released Slack Indicator profiles on CIFAR-10 with $K = 40$ extra coordinates. The profiles show that the Slack Indicator captures both near threshold behavior and the growing mass of small-norm gradients, which we use next to adapt the clipping threshold.

**Step 4: Clipping threshold adaptation.** Motivated by these two signals, we first consider the near threshold coordinate. The first coordinate $\hat{s}_{t,1}$ estimates a bin-averaged CDF value nearest the current clipping threshold $C_t$, and therefore serves as a noisy surrogate for the unclipped fraction at iteration $t$. One may naturally use this estimate to update the threshold following Adap-Clip (Andrew et al., 2021):

$$C_{t+1} \leftarrow C_t \exp\Big(\eta\big(\gamma - \hat{s}_{t,1}\big)\Big), \tag{12}$$

where $\eta$ is the adaptation step size and $\gamma$ denotes the target CDF level, equivalently a surrogate for the desired unclipped fraction. Adap-Clip sets $\gamma = 0.5$ (the median), a choice validated through extensive empirical evaluation and shown to perform robustly across tasks without hyperparameter tuning. We refer to this variant as *SlaClip-Q*. Unlike Adap-Clip, which relies on additional private queries to estimate the fraction, SlaClip-Q introduces no private query beyond the main DP-SGD release.

We further examine this adaptation rule and note a limitation. During DP-SGD training, small-norm gradients contribute little to the model update, as their effect can be dominated by the injected Gaussian noise, whereas Eq. (12) treats all gradients as equally informative. Intuitively, such gradients should not influence the threshold update. While no existing work explores this direction, the Slack Indicator offers a principled solution: by construction, the last coordinate $\hat{s}_{t,K}$ captures the CDF mass near zero and thus serves as a noisy surrogate for the mass of small-norm gradients. Appendix C provides its CDF area interpretation and DP noise scale. We therefore use the threshold adjusted signal $\hat{s}_{t,K}/C_t$ to form the dynamic target clipping ratio. This yields a dynamic target clipping ratio $\gamma_t \triangleq \Pi_{[0,1]}(1 - (1 - \hat{s}_{t,K}/C_t)/2)$. Substituting $\gamma_t$ into Eq. (12) gives the *SlaClip* adaptation:

$$C_{t+1} \leftarrow C_t \exp(\eta(\gamma_t - \hat{s}_{t,1})). \tag{13}$$

The updated threshold $C_{t+1}$ is then used for clipping in iteration $t + 1$.

---

**Algorithm 1** SlaClip (iteration $t$): SlaClip release and threshold adaptation

---

**Input:** minibatch $\mathcal{B}_t$, normalization constant $B$, clipping threshold $C_t$, extra coordinates $K$, noise multiplier $\sigma$, stepsize $\eta$
**Output:** released $\widetilde{\mathbf{g}}_t, \widetilde{\mathbf{s}}_t$, updated threshold $C_{t+1}$
Set $\lambda \leftarrow C_t/\sqrt{K}$
**for** each $i \in \mathcal{B}_t$ **do**
    Compute per-sample gradient $\mathbf{g}_{t,i} \in \mathbb{R}^d$
    Construct extended gradient $\mathbf{g}_{t,i}^+$ by (6)
**end for**
Sample $\mathcal{N}_t \sim \mathcal{N}(\mathbf{0}, (\sigma C_t/B)^2 \mathbf{I}_{d+K})$
Release $\widetilde{\mathbf{g}}_t^+ \leftarrow \frac{1}{B}\sum_{i\in\mathcal{B}_t} \mathbf{g}_{t,i}^+ + \mathcal{N}_t$
Parse $\widetilde{\mathbf{g}}_t^+ = [\widetilde{\mathbf{g}}_t; \widetilde{\mathbf{s}}_t]$ and set $\hat{\mathbf{s}}_t \leftarrow \widetilde{\mathbf{s}}_t/\lambda$
Compute $\gamma_t \leftarrow \Pi_{[0,1]}\big(1 - \big(1 - \hat{s}_{t,K}/C_t\big)/2\big)$
Update $C_{t+1} \leftarrow C_t \exp(\eta(\gamma_t - \hat{s}_{t,1}))$

---

Throughout Steps I–IV, *SlaClip* updates the clipping threshold without introducing any private query beyond the main DP-SGD release and without altering the vanilla DP-SGD Steps I-IV (Section 2); the only change on DP-SGD is the value of the clipping threshold, thereby making *SlaClip* a plug-and-play, single-release adaptive clipping method for vanilla DP-SGD, as summarized in Algorithm 1.

**Choosing $K$.** The adaptation step size $\eta$ is a standard hyperparameter in adaptive clipping, while the slack vector dimension $K$ is the only additional hyperparameter introduced by *SlaClip* when applied to DP-SGD (Algorithm 1). Although Lemma 3.2 holds for any choice of $K$, we posit that $K$ can largely impact the quality of the CDF estimation. As illustrated in Fig. 2-C, selecting $K$ involves a trade-off: larger values yield higher resolution binned CDF estimates after aggregation, but also amplify the impact of Gaussian noise, which can dominate $\hat{s}_{t,k}$, degrade CDF estimation quality, and reduce clipping utility (Appendix Table 6). This effect is also visible in Fig. 3: when using $K = 40$ for visualization, some released Slack Indicator profiles exhibit mild monotonicity violations, where larger index coordinates occasionally exceed smaller index coordinates despite the expected non-increasing ordering of the underlying CDF estimates. We resolve this trade-off by exploiting the monotonicity of the CDF and derive an upper bound

$$K \leq (B/(2\,z_{0.995}\,\sigma))^{2/3}, \tag{14}$$

which guarantees with 99.5% confidence that Gaussian noise does not induce violations of the expected non-increasing ordering of the binned CDF estimates, i.e., $\hat{s}_{t,k} \geq \hat{s}_{t,k+1}$. A detailed discussion is provided in Appendix D. Consequently, once $B$ and $\sigma$ are fixed, $K$ is determined without additional hyperparameter tuning.

# 4. Experiments

**Datasets, models, and baselines.** We evaluate *SlaClip* on five vision and text datasets: MNIST, F-MNIST (LeCun & Cortes, 1998; Xiao et al., 2017), CIFAR-10 (Krizhevsky, 2009), IMDB sentiment (Maas et al., 2011), and Names character-level classification. Each dataset is paired with an architecture commonly used in previous DP training and clipping studies (LeCun et al., 1998; Papernot et al., 2021; Bu et al., 2020), as summarized in Table 1; additional implementation details are provided in Appendix A. We compare against four representative baselines: **Vanilla-Clip** (Abadi et al., 2016), **AutoClip** (Bu et al., 2023), **Adap-Clip** (Andrew et al., 2021), and **DC-SGD-E** (Wei et al., 2025). We additionally include **SlaClip-Q**, introduced in Step 4 of Section 3, as an ablation of the adaptation rule. Code to reproduce our experiments is available in the GitHub repository.

**Evaluation protocol.** The fixed configuration setting Reg* in Section 3 is used to analyze the mechanism and privacy accounting of *SlaClip*. For empirical comparison, we use a *fairly tuned protocol* based on a shared hyperparameter pool and validation selection, to avoid favoring any method through a manually chosen training configuration. For each method, dataset, and privacy budget, we sweep over the same hyperparameter pool and select the configuration using validation accuracy. We then retrain the selected configuration with three random seeds $\{42, 43, 44\}$ and report the resulting test accuracy as mean $\pm$ std. This protocol allows each method to use its own validation selected configuration while using the same hyperparameter search space and validation selection rule. For MNIST, F-MNIST, IMDB, and Names, we sweep $lr \in \{0.01, 0.05, 0.1, 0.2, 0.5, 1\}$, $B \in \{256, 512, 1024\}$, and $C_0 \in \{0.1, 0.5, 1, 5, 10\}$; for CIFAR-10, we use the same $lr$ and $C_0$ pools and sweep $B \in \{512, 1024, 2048\}$. We additionally consider constant and cosine learning-rate schedules. For each target privacy budget and candidate batch size, we calibrate the noise multiplier $\sigma$ under the same accountant, sampling rule, and training horizon; the calibrated values are reported in Appendix Table 2.

## 4.1. Performance Comparison

We first report the main fairly tuned comparison, followed by diagnostic and hyperparameter sensitivity analyses.

**Main fairly tuned comparison.** Table 1 reports the main fairly tuned comparison. Across datasets and privacy budgets, *SlaClip* achieves competitive or improved utility, and frequently attains the best or second-best private accuracy. *SlaClip-Q* is often comparable to Adap-Clip and can outperform it in several settings, showing that the CDF information near the threshold is already useful for adapting $C_t$. The full *SlaClip* further uses the near-zero part of the CDF to reduce

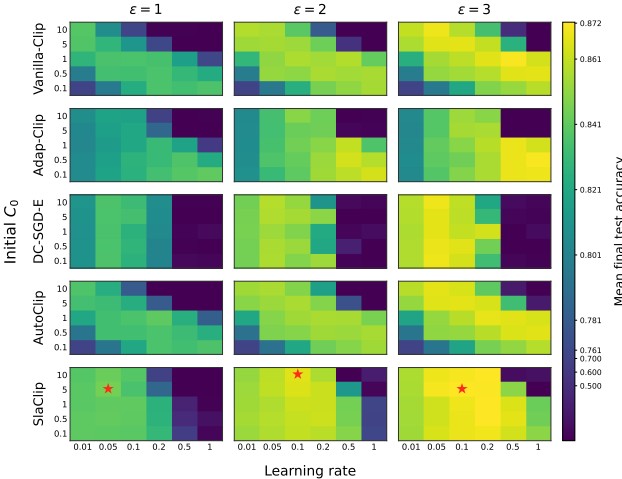

F-MNIST Heatmaps

*Figure 4.* Representative grid-search heatmaps on F-MNIST under different target privacy budgets. Rows correspond to clipping methods and columns correspond to target privacy budgets $\varepsilon \in \{1, 2, 3\}$. Within each panel, the $x$-axis is the learning rate $lr \in \{0.01, 0.05, 0.1, 0.2, 0.5, 1\}$ and the $y$-axis is the initial clipping threshold $C_0 \in \{0.1, 0.5, 1, 5, 10\}$. Each cell reports the best final test accuracy over the batch-size pool $B \in \{256, 512, 1024\}$ for the corresponding method, privacy budget, learning rate, and $C_0$. The figure illustrates how different clipping strategies respond to the interaction between learning rate and initial clipping threshold. The color scale uses a nonlinear normalization that expands the accuracy range within 0.1 of the best value and compresses lower ranges. For each privacy budget, the red star marks the best configuration across all methods and displayed hyperparameters.

the influence of small-norm gradients when forming the target clipping level, which is consistent with its improvements over *SlaClip-Q* in many settings. The Non-DP column is included only as a reference for the remaining utility gap under privacy.

**Grid-search landscape.** Figure 4 visualizes the F-MNIST grid-search landscape under the same fairly tuned protocol. This representative heatmap is included to illustrate how different clipping strategies respond to the interaction between learning rate and the initial clipping threshold $C_0$. Compared with the baselines, *SlaClip* exhibits a broader high accuracy region around the common learning-rate range near 0.1 and is less sensitive to the initial choice of $C_0$ in this region. This suggests that the Slack Indicator can help stabilize clipping threshold adaptation across a range of plausible initial thresholds, rather than requiring a narrowly tuned $C_0$.

**Controlled diagnostics.** The strongest privacy regimes require more care because the Slack Indicator needs enough updates to stabilize. In the main comparison, the target privacy budget is fixed before training, and $\sigma$ is calibrated for the full prescribed training horizon. This protocol, which calibrates $\sigma$ for the target privacy budget, differs from an

*Table 1.* **Fairly tuned comparison under matched privacy budgets.** For each method, dataset, and privacy budget, we first perform a grid search over a shared hyperparameter pool and select the configuration using validation accuracy from a single selection seed. The selected configuration is then retrained with three random seeds $\{42, 43, 44\}$, and we report the resulting test accuracy (%) as mean $\pm$ std. Unless otherwise specified, the shared search space includes learning rate $\{0.01, 0.05, 0.1, 0.2, 0.5, 1\}$, batch size $\{256, 512, 1024\}$, initial clipping threshold $C_0 \in \{0.1, 0.5, 1, 5, 10\}$, and learning-rate schedule $\{\texttt{constant}, \texttt{cos}\}$. For CIFAR-10, the batch-size pool is instead $\{512, 1024, 2048\}$. Method-specific adaptive parameters are additionally swept when applicable. The **Non-DP** column reports the corresponding non-private reference using the same model family. Within each dataset and privacy budget, the best private result is in **bold** and the second best is underlined. The selection protocol and implementation details are summarized in Appendix A.

| DATASET | MODEL | $\varepsilon$ | VANILLA-CLIP | ADAP-CLIP | DC-SGD-E | AUTOCLIP | SLACLIP-Q | SLACLIP | NON-DP |
|---|---|---|---|---|---|---|---|---|---|
| CIFAR-10 | CNN-4 | 4 | 60.24±0.19 | 59.56±0.48 | 53.22±0.43 | **60.95±0.65** | 59.52±0.35 | 60.41±0.64 | |
| | | 6 | 65.48±0.21 | 65.31±0.35 | 59.76±0.34 | 65.55±0.39 | 65.47±0.33 | **65.67±0.47** | 79.70±0.21 |
| | | 8 | 68.08±0.12 | 68.75±0.59 | 64.10±0.38 | 68.29±0.25 | 69.43±0.27 | **69.48±0.52** | |
| MNIST | CNN-2 | 1 | 92.23±0.37 | 92.67±0.34 | 93.40±0.40 | 92.61±0.31 | 92.05±0.39 | **94.15±0.32** | |
| | | 2 | 95.76±0.16 | 96.28±0.25 | 95.70±0.25 | 95.68±0.34 | 96.21±0.30 | **96.49±0.19** | 99.23±0.13 |
| | | 3 | 96.57±0.09 | 97.33±0.17 | 96.49±0.18 | 96.40±0.19 | 97.38±0.18 | **97.48±0.15** | |
| F-MNIST | CNN-2 | 1 | 83.83±0.45 | 84.11±0.36 | 83.66±0.81 | 84.05±0.55 | 83.87±0.31 | **84.56±0.36** | |
| | | 2 | 85.77±0.34 | 86.39±0.26 | 85.81±0.58 | 85.60±0.45 | 86.40±0.36 | **86.63±0.38** | 92.64±0.08 |
| | | 3 | 87.12±0.29 | 87.01±0.20 | 86.90±0.39 | 86.74±0.21 | 87.20±0.27 | **87.24±0.22** | |
| IMDB | MLP | 2 | 61.62±0.38 | 73.44±0.46 | 62.69±0.43 | **75.51±0.39** | 73.42±0.65 | 74.64±0.57 | |
| | | 4 | 67.78±0.35 | 77.12±0.53 | 71.74±0.49 | 78.54±0.47 | 77.27±0.58 | **78.62±0.50** | 83.03±0.12 |
| | | 6 | 71.04±0.23 | 78.64±0.45 | 74.60±0.37 | 79.44±0.26 | 78.96±0.34 | **79.59±0.35** | |
| NAMES | CRNN | 1 | 72.80±0.48 | **73.19±0.54** | 71.35±0.68 | 72.45±0.67 | 72.97±0.58 | 73.15±0.53 | |
| | | 2 | 75.44±0.59 | 75.54±0.36 | 75.24±0.64 | 74.34±0.60 | 76.19±0.45 | **76.48±0.41** | 84.43±0.24 |
| | | 3 | 76.63±0.45 | 76.08±0.32 | 76.58±0.58 | 75.78±0.33 | 76.36±0.37 | **76.83±0.52** | |

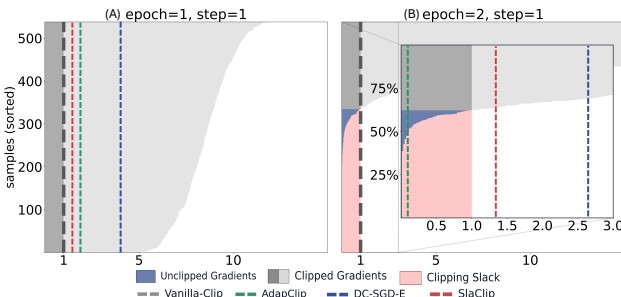

Per-sample Gradient Norms (Sorted)

*Figure 5.* Sorted per-sample gradient norms on MNIST under Vanilla DP-SGD with a fixed clipping threshold $C_t \equiv 1$, with batch size 512. (A) shows an early minibatch, where most gradient norms exceed the reference threshold and the slack signal below the threshold is sparse. (B) shows a later minibatch, where a substantial mass of small-norm gradients appears. Dashed marker lines are approximate and are provided only for visual reference.

therefore are not equivalent to training with a noise multiplier calibrated for a full $\varepsilon = 1$ training horizon. In that early-checkpoint regime, training has only just begun, so the Slack Indicator has little opportunity to track the distribution of gradient norms. Figure 5 further illustrates this behavior: early in training, most norms exceed the threshold and slack information below the threshold is sparse, whereas later a substantial near-zero mass appears. This later regime is where *SlaClip* differs most from Adap-Clip, because it uses both the coordinate near the threshold and the near-zero coordinate of the CDF profile. Figure 6 complements this analysis by showing that larger initial thresholds can help under strong privacy constraints on MNIST and F-MNIST, while *SlaClip* remains relatively stable across tested initializations on the other datasets.

**Additional controlled ablations and diagnostics.** Additional controlled ablations on $\eta$, $K$, batch size, and $C_0$ are provided in Appendix F. The step size ablation supports the default choice $\eta = 0.2$, while the $K$ ablation validates the design rule in Eq. (14). Appendix F further reports a controlled fixed recipe comparison and clipping threshold trajectories, which isolate the behavior of different clipping rules when all methods share the same manually fixed DP-SGD recipe. Together, these supplementary results support the default choices used in the main comparison and further characterize the regimes in which the Slack Indicator is most informative.

early-checkpoint diagnostic protocol, which fixes a larger final privacy budget and inspects the model before the accumulated privacy loss reaches a smaller value. Appendix F follows this latter diagnostic protocol: it inspects early checkpoints from a run targeting a larger final privacy budget $\varepsilon = 3$ before accumulated privacy loss reaches $\varepsilon = 1$. Although these checkpoints also satisfy the smaller privacy budget, they occur after only a small number of updates and

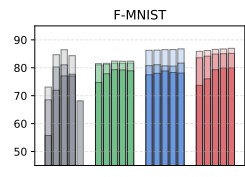
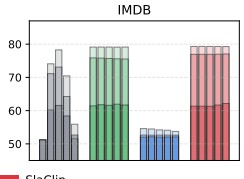
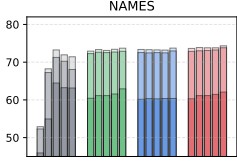

*Figure 6.* Controlled fixed recipe sensitivity to the initial clipping threshold $C_0$. For each clipping strategy, bars are ordered from left to right by $C_0 \in \{0.1, 1, 5, 10, 20\}$. Each bar is segmented by three privacy budgets per dataset, where lighter color indicates larger $\varepsilon$. Heights report test accuracy. This controlled analysis complements the grid-search heatmap by varying $C_0$ across datasets while keeping the remaining training recipe fixed.

## 5. Related Work

Due to space constraints and because we discuss and compare closely related methods alongside our methodological exposition, we defer a full review to Appendix E. Existing adaptive clipping approaches can be broadly grouped into two lines: methods such as Adap-Clip that adapt $C_t$ using additional private queries (e.g., quantile or distribution estimation), thereby requiring additional privacy accounting and requiring stronger noise to maintain a fixed privacy budget (Andrew et al., 2021; Wei et al., 2025); and methods that avoid extra queries but modify the optimization procedure, such as AutoClip and GeoClip, which introduce gradient transformations and shift sensitivity to other optimization choices; moreover, both methods rely on pretraining when evaluated specific datasets (Bu et al., 2023; Gilani et al., 2025; Xia et al., 2023).

## 6. Conclusion

We propose *SlaClip*, a single-release adaptive clipping, plug-and-play method for vanilla DP-SGD. *SlaClip* operates entirely within the standard DP-SGD release and without introducing additional private queries or optimization components. This is enabled by a tailored Slack Indicator that encodes slack information into extended gradients while preserving the original global $\ell_2$ sensitivity of the query. Experimental results show that *SlaClip* improves model utility over existing baselines in most settings. *SlaClip* is modular and composes naturally with other DP-SGD mechanisms; for example, under per-layer clipping (McMahan et al., 2018), *SlaClip* can be applied in parallel via layer-wise Slack Indicators. Finally, *SlaClip* currently exploits only the most informative Slack Indicator coordinates, leaving exploitation of its structure as a promising direction for future work.

## Acknowledgements

Shuyan Zou is supported by the ECS scholarship from the School of Electronics and Computer Science, University of Southampton. Shaowei Wang is supported by National Natural Science Foundation of China (No.62372120, No.62102108), Guangdong Provincial Association for Science and Technology (No.SKXRC2025407), and Guangzhou Basic and Applied Basic Research Foundation (No.2025A03J3182).

## Impact Statement

This paper presents work whose goal is to advance the field of Machine Learning. There are many potential societal consequences of our work, none which we feel must be specifically highlighted here.

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

# A. Implementation Details and Hyperparameters

**DP training pipeline.**    All private experiments are implemented in Opacus (Yousefpour et al., 2021) and trained with DP-SGD. Privacy is tracked using the Rényi differential privacy (RDP) accountant provided by Opacus with its default set of RDP orders, and the final privacy guarantee is reported by converting the accumulated RDP to $(\varepsilon, \delta)$ at the end of training. For the main fairly tuned comparison, the noise multiplier $\sigma$ is calibrated for each target privacy budget using the same accountant, sampling rule, and training horizon. Per-sample gradients are computed using the hooks-based mechanism in Opacus. After validation-based hyperparameter selection, each selected configuration is retrained with three random seeds $\{42, 43, 44\}$, and we report mean±std test accuracy.

**Datasets and model architectures.**    We evaluate on five benchmarks: MNIST (LeCun & Cortes, 1998), F-MNIST (Xiao et al., 2017), CIFAR-10 (Krizhevsky, 2009), IMDB (Maas et al., 2011), and Names. For each dataset, we use a fixed architecture across all methods:

- **CIFAR-10: CIFAR-ConvNet (AvgPool–GAP CNN).** A 4-layer CNN composed of $3 \times 3$ convolution–ReLU blocks, average-pooling for downsampling, global average pooling, and a linear classifier.

- **MNIST/F-MNIST: MNIST-ConvNet (LeNet-style CNN).** Two convolution–ReLU–maxpool blocks followed by a two-layer MLP classifier head.

- **IMDB: IMDB-MLP (Deep Averaging Network).** Token embeddings (16-d), global average pooling over the sequence, and a two-layer MLP for binary classification.

- **Names: Char-RNN (character-level recurrent model).** Character embeddings (128-d) and a DP-LSTM encoder; the last valid timestep representation is fed into a linear classifier.

**Main fairly tuned protocol.**    For the main comparison in Table 1, all methods are evaluated using the same hyperparameter search space and validation selection rule. For each method, dataset, and target privacy budget, we select one configuration using validation accuracy from a single selection seed, and then retrain the selected configuration with seeds $\{42, 43, 44\}$. For MNIST, F-MNIST, IMDB, and Names, we sweep $lr \in \{0.01, 0.05, 0.1, 0.2, 0.5, 1\}$, $B \in \{256, 512, 1024\}$, $C_0 \in \{0.1, 0.5, 1, 5, 10\}$. For CIFAR-10, we use the same learning rate and $C_0$ pools and sweep $B \in \{512, 1024, 2048\}$. We additionally consider constant and cosine learning-rate schedules. We train CIFAR-10 and IMDB for 90 epochs, and MNIST, F-MNIST, and Names for 30 epochs. For each target privacy budget, the noise multiplier $\sigma$ is calibrated separately for each dataset, batch size, training horizon, sampling rule, and RDP accountant. Table 2 reports the calibrated values used in the grid search. When applicable, method specific adaptive parameters are swept within the same validation selection protocol. Because the full list of selected configurations is large, we provide the sample command line arguments with selected hyperparameters in the released code repository for reproducibility.

**Privacy parameters.**    Unless otherwise stated, we use $\delta = 10^{-5}$ for MNIST, F-MNIST, CIFAR-10, and IMDB. For Names, we use $\delta = 8 \times 10^{-5}$ to match its sample size and evaluation protocol. For each target $\varepsilon$, the noise multiplier $\sigma$ is calibrated under the same accountant and sampling rule for all methods being compared. Thus, methods are compared at matched target privacy budgets while allowing validation selected training configurations under the same hyperparameter search space and selection rule.

**Controlled fixed configuration experiments.**    Some appendix experiments use a controlled fixed configuration setting to isolate specific mechanisms, such as early stage behavior, sensitivity to the initial clipping threshold, or sensitivity to the Slack Indicator dimension $K$. Those experiments are not part of the main fairly tuned comparison. Unless otherwise stated in the corresponding appendix section, they use the default fixed recipe specified below.

**Default fixed recipe for controlled appendix experiments.**    Unless otherwise stated, controlled appendix experiments use the following fixed DP-SGD recipe. For MNIST, F-MNIST, CIFAR-10, and IMDB, all methods use SGD with learning rate 0.1, momentum 0.9, weight decay $5 \times 10^{-4}$, and a cosine learning-rate schedule. We use batch size $B = 1024$ for CIFAR-10, $B = 512$ for MNIST and F-MNIST, and $B = 256$ for IMDB. For Names, we use a constant learning-rate configuration with learning rate 2.0, momentum 0, weight decay 0, batch size $B = 512$, and a constant schedule. Unless a section explicitly sweeps $C_0$, we use the common initial clipping threshold $C_0 = 1$ for all methods. Unless a section

*Table 2.* Noise calibration details for the main fairly tuned comparison. For each dataset, candidate batch size $B$, and target privacy budget $\varepsilon$, we calibrate the noise multiplier $\sigma$ using the same RDP accountant, sampling rule, and training horizon. We use the effective accountant sampling rate $q_{\text{eff}} = 1/\lceil N/B \rceil$ and steps $= \lceil N/B \rceil \times$ epochs. For MNIST, F-MNIST, CIFAR-10, and IMDB, we use $\delta = 10^{-5}$; for Names, we use $\delta = 8 \times 10^{-5}$. Values of $\sigma$ are rounded to three decimals.

| Dataset | $N$ | Epochs | $B$ | $q_{\text{eff}}$ | Steps | $(\varepsilon, \sigma)$ |
|---|---|---|---|---|---|---|
| CIFAR-10 | 50000 | 90 | 512 | 0.01020 | $98 \times 90 = 8820$ | $(4, 1.317), (6, 1.035), (8, 0.897)$ |
| | | | 1024 | 0.02041 | $49 \times 90 = 4410$ | $(4, 1.739), (6, 1.317), (8, 1.109)$ |
| | | | 2048 | 0.04000 | $25 \times 90 = 2250$ | $(4, 2.340), (6, 1.723), (8, 1.416)$ |
| MNIST | 60000 | 30 | 256 | 0.00426 | $235 \times 30 = 7050$ | $(1, 1.624), (2, 1.034), (3, 0.856)$ |
| | | | 512 | 0.00847 | $118 \times 30 = 3540$ | $(1, 2.189), (2, 1.309), (3, 1.031)$ |
| | | | 1024 | 0.01695 | $59 \times 30 = 1770$ | $(1, 3.015), (2, 1.725), (3, 1.304)$ |
| F-MNIST | 60000 | 30 | 256 | 0.00426 | $235 \times 30 = 7050$ | $(1, 1.624), (2, 1.034), (3, 0.856)$ |
| | | | 512 | 0.00847 | $118 \times 30 = 3540$ | $(1, 2.189), (2, 1.309), (3, 1.031)$ |
| | | | 1024 | 0.01695 | $59 \times 30 = 1770$ | $(1, 3.015), (2, 1.725), (3, 1.304)$ |
| IMDB | 25000 | 90 | 256 | 0.01020 | $98 \times 90 = 8820$ | $(2, 2.194), (4, 1.317), (6, 1.035)$ |
| | | | 512 | 0.02041 | $49 \times 90 = 4410$ | $(2, 3.024), (4, 1.739), (6, 1.317)$ |
| | | | 1024 | 0.04000 | $25 \times 90 = 2250$ | $(2, 4.176), (4, 2.340), (6, 1.723)$ |
| Names | 18067 | 30 | 256 | 0.01408 | $71 \times 30 = 2130$ | $(1, 2.455), (2, 1.457), (3, 1.132)$ |
| | | | 512 | 0.02778 | $36 \times 30 = 1080$ | $(1, 3.377), (2, 1.933), (3, 1.451)$ |
| | | | 1024 | 0.05556 | $18 \times 30 = 540$ | $(1, 4.722), (2, 2.641), (3, 1.932)$ |

explicitly varies $K$, we choose $K$ according to the rule in Appendix D; for *SlaClip*, we use $\eta = 0.5$ in these controlled fixed recipe experiments. The noise multiplier is fixed to $\sigma = 1.0$, and privacy is tracked with the same RDP accountant as in the main experiments. When an ablation varies one quantity, such as $C_0$, $K$, or $B$, all other settings follow this fixed recipe.

**Codebase notes.** All experiments are launched from a unified entry point (`run_exp.py`). We implement SlaClip and all baselines as optimizer variants under the Opacus (Yousefpour et al., 2021) optimizer interface, so the DP training pipeline and privacy accounting are shared across methods. Across methods, the differences are in the clipping rule, threshold adaptation rule, and validation selected hyperparameters under the protocol described above.

**Environment.** All experiments are run on a single NVIDIA RTX 4090 GPU (24GB). We use Python 3.10 and PyTorch with CUDA support; all dependencies are specified in the provided environment configuration.

## B. Privacy Analysis of SlaClip

This appendix provides the complete derivation behind Theorem 3.1 and its application to the *SlaClip* release in $\mathbb{R}^{d+K}$ (Eq. (9)). We first show that replacing a Gaussian query by a sensitivity-preserving extension does not change the RDP guarantee of the Gaussian mechanism. We then apply this result to the extended average query $f_{\text{avg}}^{+}$ used by *SlaClip*. Since *SlaClip* uses the same subsampling rule, the same noise multiplier, and preserves the original global $\ell_2$ sensitivity of vanilla DP-SGD, both methods are accounted with the same per-step privacy cost upper bound under the same accountant; below we instantiate this argument with RDP.

**Complete proof of Theorem 3.1.** For completeness, we restate the Gaussian-layer claim in the notation of Theorem 3.1 and provide the full derivation.

**Lemma B.1.** *Let $f : \mathcal{D} \to \mathbb{R}^d$ and $f^+ : \mathcal{D} \to \mathbb{R}^{d+K}$ be deterministic query functions on $\mathcal{D}$, and let $D \sim D'$ denote adjacent datasets. Let $D_\alpha(* \| *)$ denote the Rényi divergence with order $\alpha$. If*

$$\Delta_2(f) = \Delta_2(f^+) = \Delta,$$

*then*

$$\sup_{D \sim D'} D_\alpha\big(\mathcal{N}(f(D), (\sigma\Delta)^2 \mathbf{I}_d) \,\big\|\, \mathcal{N}(f(D'), (\sigma\Delta)^2 \mathbf{I}_d)\big) = \sup_{D \sim D'} D_\alpha\big(\mathcal{N}(f^+(D), (\sigma\Delta)^2 \mathbf{I}_{d+K}) \,\big\|\, \mathcal{N}(f^+(D'), (\sigma\Delta)^2 \mathbf{I}_{d+K})\big).$$

*Moreover, both quantities equal $\alpha/(2\sigma^2)$.*

*Proof.* By the exact closed form of the order-$\alpha$ Rényi divergence between Gaussian distributions with matched isotropic covariance (Mironov, 2017),

$$D_\alpha\big(\mathcal{N}(\mu, \tau^2 \mathbf{I}_m) \big\| \mathcal{N}(\mu', \tau^2 \mathbf{I}_m)\big) = \frac{\alpha}{2\tau^2} \|\mu - \mu'\|^2.$$

Applying this formula with $\tau = \sigma\Delta$ gives, for every adjacent pair $D \sim D'$,

$$D_\alpha\big(\mathcal{N}(f(D), (\sigma\Delta)^2 \mathbf{I}_d) \big\| \mathcal{N}(f(D'), (\sigma\Delta)^2 \mathbf{I}_d)\big) = \frac{\alpha}{2\sigma^2\Delta^2} \|f(D) - f(D')\|^2. \tag{15}$$

Likewise,

$$D_\alpha\big(\mathcal{N}(f^+(D), (\sigma\Delta)^2 \mathbf{I}_{d+K}) \big\| \mathcal{N}(f^+(D'), (\sigma\Delta)^2 \mathbf{I}_{d+K})\big) = \frac{\alpha}{2\sigma^2\Delta^2} \|f^+(D) - f^+(D')\|^2. \tag{16}$$

Taking the supremum over adjacent datasets in Eq. (15), we obtain

$$
\begin{aligned}
\sup_{D \sim D'} D_\alpha\big(\mathcal{N}(f(D), (\sigma\Delta)^2 \mathbf{I}_d) \big\| \mathcal{N}(f(D'), (\sigma\Delta)^2 \mathbf{I}_d)\big) &= \frac{\alpha}{2\sigma^2\Delta^2} \sup_{D \sim D'} \|f(D) - f(D')\|^2 \\
&= \frac{\alpha}{2\sigma^2\Delta^2} \Delta_2(f)^2 \\
&= \frac{\alpha}{2\sigma^2\Delta^2} \Delta^2 \\
&= \frac{\alpha}{2\sigma^2}.
\end{aligned}
\tag{17}
$$

Similarly, from Eq. (16),

$$
\begin{aligned}
\sup_{D \sim D'} D_\alpha\big(\mathcal{N}(f^+(D), (\sigma\Delta)^2 \mathbf{I}_{d+K}) \big\| \mathcal{N}(f^+(D'), (\sigma\Delta)^2 \mathbf{I}_{d+K})\big) &= \frac{\alpha}{2\sigma^2\Delta^2} \sup_{D \sim D'} \|f^+(D) - f^+(D')\|^2 \\
&= \frac{\alpha}{2\sigma^2\Delta^2} \Delta_2(f^+)^2 \\
&= \frac{\alpha}{2\sigma^2\Delta^2} \Delta^2 \\
&= \frac{\alpha}{2\sigma^2}.
\end{aligned}
\tag{18}
$$

Therefore,

$$\sup_{D \sim D'} D_\alpha\big(\mathcal{N}(f(D), (\sigma\Delta)^2 \mathbf{I}_d) \big\| \mathcal{N}(f(D'), (\sigma\Delta)^2 \mathbf{I}_d)\big) = \sup_{D \sim D'} D_\alpha\big(\mathcal{N}(f^+(D), (\sigma\Delta)^2 \mathbf{I}_{d+K}) \big\| \mathcal{N}(f^+(D'), (\sigma\Delta)^2 \mathbf{I}_{d+K})\big),$$

which proves the claim. $\qquad\square$

**Application to SlaClip.** Within one DP-SGD iteration, the vanilla average query is

$$f_{\text{avg}}(B_t) = \frac{1}{B} \sum_{i \in B_t} \text{Clip}_{C_t}(g_{t,i}),$$

whereas SlaClip releases the extended average query

$$f^+_{\text{avg}}(B_t) = \frac{1}{B} \sum_{i \in B_t} g^+_{t,i}.$$

By Lemma B.2, these two queries preserve the same original global $\ell_2$ sensitivity under add/remove adjacency:

$$\Delta_2(f_{\text{avg}}) = \Delta_2(f^+_{\text{avg}}) = \frac{C_t}{B}.$$

Therefore, by Theorem 3.1 (equivalently, Lemma B.1), replacing the vanilla Gaussian release by the extended release in Eq. (9) preserves the RDP guarantee of the Gaussian mechanism.

Moreover, SlaClip uses the same subsampling rule, the same noise multiplier $\sigma$, and preserves the original global $\ell_2$ sensitivity $C_t/B$ of vanilla DP-SGD. Hence, under the same accountant in Reg*, SlaClip and vanilla DP-SGD yield the same per-step RDP guarantee.

**Lemma B.2** (Per-sample $\ell_2$ bound and sensitivity of the average query). *For all $i \in \mathcal{B}_t$, the extended gradient satisfies* $\|\mathbf{g}_{t,i}^+\| \le C_t$. *Under add/remove adjacency between sampled minibatches, using the fixed normalization constant $B$, the average query $f_{avg}^+(\mathcal{B}) = \frac{1}{B} \sum_{i \in \mathcal{B}} \mathbf{g}_{t,i}^+$ has $\ell_2$ sensitivity $\Delta_2(f_{avg}^+) = C_t/B$.*

*Proof.* We proceed in two parts.

**Part I: Per-sample $\ell_2$ bound.** Fix any $i \in \mathcal{B}_t$ and consider two cases.

*Case 1:* $\|\mathbf{g}_{t,i}\| > C_t$. By Eq. (6), the extended vector equals the clipped $d$-dimensional gradient concatenated with a zero slack vector, hence

$$\mathbf{g}_{t,i}^+ = [C_t \cdot \mathbf{g}_{t,i}/\|\mathbf{g}_{t,i}\|; \ \mathbf{0}].$$

Therefore,

$$\|\mathbf{g}_{t,i}^+\|^2 = \left\| C_t \frac{\mathbf{g}_{t,i}}{\|\mathbf{g}_{t,i}\|} \right\|^2 + \|\mathbf{0}\|^2 = C_t^2,$$

and therefore $\|\mathbf{g}_{t,i}^+\| = C_t$.

*Case 2:* $\|\mathbf{g}_{t,i}\| \le C_t$. By Eq. (7), the slack vector $\mathbf{s}_{t,i}$ is constructed so that

$$\|\mathbf{s}_{t,i}\|^2 = a\lambda^2 + b^2 \le a\lambda^2 + \lambda b = \lambda(a\lambda + b) = \lambda \cdot \sqrt{K} \cdot (C_t - \|\mathbf{g}_{t,i}\|) \le C_t(C_t - \|\mathbf{g}_{t,i}\|),$$

where the last inequality uses $\lambda = C_t/\sqrt{K} \le C_t$. Since the extension is a concatenation, the squared norm decomposes as

$$\|\mathbf{g}_{t,i}^+\|^2 = \|\mathbf{g}_{t,i}\|^2 + \|\mathbf{s}_{t,i}\|^2.$$

Combining the above bounds yields

$$\|\mathbf{g}_{t,i}^+\|^2 \le \|\mathbf{g}_{t,i}\|^2 + C_t(C_t - \|\mathbf{g}_{t,i}\|) = \left(\|\mathbf{g}_{t,i}\|^2 - C_t\|\mathbf{g}_{t,i}\|\right) + C_t^2 \le C_t^2,$$

because for $x = \|\mathbf{g}_{t,i}\| \in [0, C_t]$ we have $x^2 - C_t x \le 0$ (equivalently, $x(x - C_t) \le 0$). Hence $\|\mathbf{g}_{t,i}^+\| \le C_t$.

In both cases, $\|\mathbf{g}_{t,i}^+\| \le C_t$ holds for every $i$.

**Why $\lambda = C_t/\sqrt{K}$ is maximal.** We show that $\lambda \le C_t/\sqrt{K}$ is not only sufficient but also necessary to guarantee $\|\mathbf{g}_{t,i}^+\| \le C_t$ for all $i$.

By construction, each coordinate of the slack vector $\mathbf{s}_{t,i} \in \mathbb{R}^K$ is bounded in magnitude by $\lambda$. Hence,

$$\|\mathbf{s}_{t,i}\|_\infty \le \lambda \implies \|\mathbf{s}_{t,i}\|_2 \le \sqrt{K}\,\lambda,$$

where the upper bound is attained when all $K$ coordinates equal $\lambda$.

Consider the worst-case sample for the concatenated norm, namely $\|\mathbf{g}_{t,i}\| = 0$. Then, since the extension is a concatenation,

$$\|\mathbf{g}_{t,i}^+\|^2 = \|\mathbf{g}_{t,i}\|^2 + \|\mathbf{s}_{t,i}\|^2 = \|\mathbf{s}_{t,i}\|^2 \le K\lambda^2.$$

To ensure $\|\mathbf{g}_{t,i}^+\| \le C_t$ for all $i$, it is therefore necessary that

$$\sqrt{K}\,\lambda \le C_t,$$

or equivalently,

$$\lambda \le \frac{C_t}{\sqrt{K}}.$$

Thus, $\lambda = C_t/\sqrt{K}$ is the maximum admissible value that preserves the clipping constraint $\|\mathbf{g}_{t,i}^+\| \le C_t$.

**Part II: $\ell_2$ sensitivity of the average query is $C_t/B$.** Define the normalized query on a sampled minibatch $\mathcal{B}$ using the fixed normalization constant $B$ by

$$f^+_{avg}(\mathcal{B}) = \frac{1}{B} \sum_{i \in \mathcal{B}} \mathbf{g}^+_{t,i}.$$

We consider add/remove adjacency: $\mathcal{B} \sim \mathcal{B}'$ means the two minibatches differ by at most one example, i.e., one element is added or removed (but not both). Equivalently, there exists a set $S$ and an element $u$ (possibly empty) such that either

$$\mathcal{B} = S \cup \{u\}, \quad \mathcal{B}' = S \qquad \text{or} \qquad \mathcal{B} = S, \quad \mathcal{B}' = S \cup \{u\}.$$

For notational uniformity, in the removal case we treat the missing element as $u = \emptyset$ and set $\mathbf{g}(\emptyset) = \mathbf{0}$.

Then, in both cases,

$$f^+_{avg}(\mathcal{B}) - f^+_{avg}(\mathcal{B}') = \frac{1}{B} \left( \sum_{i \in S} \mathbf{g}_i + \mathbf{g}_u \right) - \frac{1}{B} \left( \sum_{i \in S} \mathbf{g}_i \right) = \frac{1}{B} \mathbf{g}_u, \tag{19}$$

where we abbreviate $\mathbf{g}_i = \mathbf{g}^+_{t,i}$. Taking norms gives

$$\left\| f^+_{avg}(\mathcal{B}) - f^+_{avg}(\mathcal{B}') \right\| = \frac{1}{B} \| \mathbf{g}_u \|.$$

By Part I, $\| \mathbf{g}^+_{t,i} \| \leq C_t$ for all $i$, and $\| \mathbf{0} \| = 0 \leq C_t$ covers $u = \emptyset$. Therefore,

$$\left\| f^+_{avg}(\mathcal{B}) - f^+_{avg}(\mathcal{B}') \right\| \leq \frac{C_t}{B}.$$

Taking the supremum over adjacent minibatches yields

$$\Delta_2(f^+_{avg}) := \sup_{\mathcal{B} \sim \mathcal{B}'} \left\| f^+_{avg}(\mathcal{B}) - f^+_{avg}(\mathcal{B}') \right\| \leq \frac{C_t}{B}.$$

Moreover, the bound is tight: choose $\mathcal{B} = S \cup \{u\}$ and $\mathcal{B}' = S$ with $\| \mathbf{g}_u \| = C_t$, so that

$$\left\| f^+_{avg}(\mathcal{B}) - f^+_{avg}(\mathcal{B}') \right\| = \frac{1}{B} \| \mathbf{g}_u \| = \frac{C_t}{B}.$$

Hence $\Delta_2(f^+_{avg}) = C_t/B$. Since the original clipped gradients also satisfy $\| \mathbf{g}_{t,i} \| \leq C_t$ for all $i$, the same argument gives $\Delta_2(f_{avg}) = C_t/B$, and therefore $\Delta_2(f^+_{avg}) = \Delta_2(f_{avg})$. $\qquad\square$

**Marginal equivalence of the first $d$ coordinates.** By construction in Eq. (6), the first $d$ coordinates of $\mathbf{g}^+_{t,i}$ coincide with the vanilla per-sample clipped gradient used by DP-SGD. Moreover, the Gaussian noise in Eq. (9) is isotropic in $\mathbb{R}^{d+K}$, hence its first $d$ coordinates are distributed as $\mathcal{N}\left(\mathbf{0}, \left(\frac{\sigma C_t}{B}\right)^2 \mathbf{I}_d\right)$. Therefore, the marginal distribution of $\widetilde{\mathbf{g}}_t$ extracted from $\widetilde{\mathbf{g}}^+_t$ matches the vanilla DP-SGD gradient release under the same $(B, \sigma, C_t)$ and the same sampling rule.

## C. Deriving the Threshold Adjusted Near-Zero Signal

This appendix derives the near-zero Slack Indicator signal used in the SlaClip update rule. Let $\widetilde{\mathbf{s}}_t$ denote the released, unnormalized slack coordinates in Eq. (10), and let

$$\hat{\mathbf{s}}_t \triangleq \widetilde{\mathbf{s}}_t / \lambda, \qquad \lambda = C_t / \sqrt{K},$$

denote the normalized Slack Indicator. SlaClip uses $\hat{s}_{t,1}$ as a near threshold CDF estimate and uses the threshold adjusted near-zero signal $\hat{s}_{t,K}/C_t$ to form the dynamic target clipping ratio.

**Step 1: Near-zero CDF area captured by the last Slack Indicator coordinate.** Let $F_t(u) \triangleq \Pr(\|\mathbf{g}_{t,i}\| \leq u)$ denote the CDF of per-sample gradient norms at iteration $t$. Recall that the $k$-th Slack Indicator coordinate corresponds to the gradient norm interval

$$[\, C_t - kC_t/K, \; C_t - (k-1)C_t/K \,].$$

Thus, the last coordinate $k = K$ corresponds to the near-zero interval $[0, C_t/K]$. From Eq. (11), the normalized released coordinate can be written as

$$\hat{s}_{t,K} = \frac{K}{BC_t} \sum_{i \in \mathcal{B}_t} \int_0^{C_t/K} \mathbf{1}\{\|\mathbf{g}_{t,i}\| \leq u\} \, du + \varepsilon_{t,K}, \tag{20}$$

where $\varepsilon_{t,K}$ denotes the normalized Gaussian noise. Ignoring the zero-mean Gaussian noise and taking the population view, we obtain

$$\mathbb{E}[\hat{s}_{t,K}] = \frac{K}{C_t} \int_0^{C_t/K} F_t(u) \, du. \tag{21}$$

Thus, $\hat{s}_{t,K}$ is a bin-averaged CDF signal over the near-zero interval $[0, C_t/K]$. It is not a point probability such as $F_t(C_t/K)$, but it increases with the concentration of small-norm gradients in this interval and is smoother than a point estimate because it averages the CDF over an interval.

**Step 2: Threshold adjustment used by SlaClip.** SlaClip further divides the last normalized coordinate by the current clipping threshold:

$$\frac{\hat{s}_{t,K}}{C_t} = \frac{K}{BC_t^2} \sum_{i \in \mathcal{B}_t} \int_0^{C_t/K} \mathbf{1}\{\|\mathbf{g}_{t,i}\| \leq u\} \, du + \frac{\varepsilon_{t,K}}{C_t}. \tag{22}$$

Equivalently, in expectation and ignoring the zero-mean noise,

$$\mathbb{E}\left[\frac{\hat{s}_{t,K}}{C_t}\right] = \frac{K}{C_t^2} \int_0^{C_t/K} F_t(u) \, du. \tag{23}$$

Equation (23) shows that $\hat{s}_{t,K}/C_t$ is a threshold adjusted near-zero CDF area signal. Compared with $\hat{s}_{t,K}$ alone, the division by $C_t$ makes the near-zero signal depend on the current clipping scale. This provides a scale aware control signal for adjusting the clipping threshold.

**Step 3: DP noise in the threshold adjusted signal.** From the single Gaussian release in Eq. (9)–(10), each unnormalized slack coordinate is perturbed by additive Gaussian noise with standard deviation $\sigma C_t/B$:

$$\widetilde{s}_{t,K} = \widehat{s}_{t,K}^{(0)} + \xi_{t,K}, \qquad \xi_{t,K} \sim \mathcal{N}\left(0, \left(\frac{\sigma C_t}{B}\right)^2\right), \tag{24}$$

where $\widehat{s}_{t,K}^{(0)}$ denotes the corresponding noise free slack coordinate. Since $\hat{s}_{t,K} = \widetilde{s}_{t,K}/\lambda$ and $\lambda = C_t/\sqrt{K}$, the normalized noise term satisfies

$$\varepsilon_{t,K} = \frac{\xi_{t,K}}{\lambda} \sim \mathcal{N}\left(0, \left(\frac{\sigma\sqrt{K}}{B}\right)^2\right). \tag{25}$$

Therefore, the noise in the threshold adjusted signal is

$$\frac{\varepsilon_{t,K}}{C_t} \sim \mathcal{N}\left(0, \left(\frac{\sigma\sqrt{K}}{BC_t}\right)^2\right). \tag{26}$$

Combining Eq. (23) and Eq. (26), the signal used in the controller can be summarized as

$$\frac{\hat{s}_{t,K}}{C_t} = \frac{K}{C_t^2} \int_0^{C_t/K} F_t(u) \, du + \mathcal{N}\left(0, \left(\frac{\sigma\sqrt{K}}{BC_t}\right)^2\right), \tag{27}$$

under the population approximation.

**Step 4: Use in the SlaClip controller.** The raw threshold adjusted signal may fall outside $[0, 1]$ because of Gaussian noise and because it is a scale adjusted CDF area proxy rather than a probability. SlaClip therefore clips it before constructing the dynamic target:

$$r_t \triangleq \Pi_{[0,1]}\left(\frac{\hat{s}_{t,K}}{C_t}\right). \tag{28}$$

The dynamic target clipping ratio is then

$$\gamma_t \triangleq \Pi_{[0,1]}\left(1 - \frac{1 - r_t}{2}\right), \tag{29}$$

and the clipping threshold is updated by

$$C_{t+1} \leftarrow C_t \exp(\eta(\gamma_t - \hat{s}_{t,1})). \tag{30}$$

Here, $\hat{s}_{t,1}$ provides the near threshold CDF feedback, while $r_t$ provides a threshold adjusted proxy for the prevalence of very small gradients. This matches the update rule used in Algorithm 1.

## D. Selecting $K$

This appendix provides a practical recipe for choosing the number of extension dimensions $K$ by balancing (i) the *resolution* along the gradient norm (equivalently, slack) axis and (ii) the *noise* on the released *Slack Indicator* $\hat{\mathbf{s}}_t$. Throughout, we keep the same notation as the main text: the per-slot scale is denoted by $\lambda$, and we consider the choice:

$$\lambda = \frac{C_t}{\sqrt{K}},$$

The gradient norm bins represented by the Slack Indicator have endpoints $C_t - k \cdot C_t/K$ for $k = 0, \dots, K$, while $\lambda = C_t/\sqrt{K}$ is the coordinate wise slack scale used for encoding and normalization.

**Setup and notation.** At iteration $t$, let $\mathbf{g}_{t,i} \in \mathbb{R}^d$ be the per-sample gradient and let $C_t$ be the clipping threshold. Define the (nonnegative) clipping slack value as

$$\left[C_t - \|\mathbf{g}_{t,i}\|\right]_+ \in [0, C_t]. \tag{31}$$

### D.1. Explicit noise forms and an SNR-based upper bound for $K$ (99% confidence)

**(A) Noise in $\hat{s}_{t,k} = \widetilde{s}_{t,k}/\lambda$.** Step 1 releases the averaged $(d+K)$-dimensional vector in Eq. (9) with isotropic Gaussian noise whose coordinate wise variance is $(\sigma C_t/B)^2$. Consequently, each released slack coordinate $s_{t,k}$ inherits additive Gaussian noise, and after normalization by $\lambda$ we have

$$\hat{s}_{t,k} = (\text{signal term}) + \mathcal{N}\left(0, \left(\frac{\sigma C_t}{B\lambda}\right)^2\right), \qquad k = 1, \dots, K, \tag{32}$$

where the "signal term" denotes the same expression with the Gaussian noise removed.

**(B) Explicit noise forms: $\lambda = C_t/\sqrt{K}$.** Plugging the $\lambda$ choice into (32) yields:

$$\lambda = \frac{C_t}{\sqrt{K}} : \quad \hat{s}_{t,k} = (\text{signal term}) + \mathcal{N}\left(0, \left(\frac{\sigma\sqrt{K}}{B}\right)^2\right). \tag{33}$$

**(C) 99% CI half-widths.** From (33), the (marginal) 99% confidence half-width is

$$\text{HW}_{99}^{(\sqrt{K})} = z_{0.995} \cdot \frac{\sigma\sqrt{K}}{B}, \tag{34}$$

where $z_{0.995} \approx 2.576$. *Remark.* (34) are per-coordinate (marginal) half-width. A simultaneous confidence band over all $k = 1, \dots, K$ can be obtained by replacing $z_{0.995}$ with $z_{1-\alpha/(2K)}$ (Bonferroni).

*Table 3.* 99% marginal SNR-based upper bounds and practical choices of $K$ for $\sigma = 1$. Bounds are computed from Eq. (36) with $z_{0.995} = 2.576$. The practical choices are convenient nearby values below the corresponding bound.

| $B$ | $K_{\mathrm{max,99}}$ | Practical $K$ |
|---|---|---|
| 128 | 8.51 | 8 |
| 256 | 13.52 | 10 |
| 512 | 21.46 | 20 |
| 1024 | 34.06 | 30 |
| 2048 | 54.06 | 50 |

| Method | Extra private query | Significant Extra Compute | Requires pretraining / public / auxiliary resources | Modification to Vanilla DP-SGD Gradients |
|---|---|---|---|---|
| SlaClip | No | No | No | No; Preserves the vanilla clipped gradient update while adapting the clipping threshold through the same DP-SGD Gaussian release |
| Adap-Clip | Yes (private clipped-count) | No | No | Yes; Applying an additional private statistic |
| DC-SGD | Yes (private norm-distribution estimation) | No | No | Yes; Applying several additional private statistics |
| AutoClip | No | No | Yes; pretraining needed for different datasets | Yes; Modifies vanilla DP-SGD through norm normalized per-example clipping |
| DP-PSAC | No | Yes; still depends on method-specific parameter tuning | Yes; pretraining needed for different datasets | Yes; Modifies vanilla DP-SGD through per-sample adaptive clipping / weighting rules |
| GeoClip | No | Yes; introduces nontrivial geometry / estimation hyperparameters and calibration burden | Yes; pretraining needed for different datasets | Yes; Modifies vanilla DP-SGD by performing clipping / noising in a transformed geometry-aware space and mapping back |

*Table 4.* Method-level comparison of direct baselines relative to vanilla DP-SGD. The table highlights whether a method introduces additional private estimation, extra tuning or calibration burden, dependence on external resources, and whether it modifies the vanilla DP-SGD gradient pipeline.

**(D) An SNR-based upper bound for $K$ and practical recommendations (99%).** To prevent *Slack Indicator* noise from dominating the effective resolution across $K$ slots, we adopt a conservative adjacent-resolution design rule: require the 99% noise half-width to be no larger than a constant multiple of the typical adjacent-slot scale $O(1/K)$ in the normalized domain. Concretely, we impose

$$\mathrm{HW}_{99} \ \leq \ \frac{1}{2K}. \tag{35}$$

**For $\lambda = C_t/\sqrt{K}$,** substituting (34) into (35) gives

$$K \ \leq \ K_{\mathrm{max,99}}^{(\sqrt{K})} \triangleq \left( \frac{B}{2\, z_{0.995}\, \sigma} \right)^{2/3}. \tag{36}$$

Since larger $K$ improves resolution, a practical choice is to take the largest $K$ that satisfies the above SNR upper bound, and then choose a convenient nearby value below the bound.

**(E) Numerical guidelines for $\sigma = 1$.** Table 3 reports the resulting $K_{\mathrm{max,99}}$ values and practical choices of $K$ for representative nominal batch sizes $B \in \{128, 256, 512, 1024, 2048\}$ under $\sigma = 1$. These values are intended as default choices rather than additionally tuned hyperparameters.

## E. Detailed Related Work

**DP-SGD and privacy accounting.** DP-SGD clips per-sample gradients and adds Gaussian noise calibrated to the clipping threshold (Abadi et al., 2016). Practical privacy accounting is commonly performed using Rényi differential privacy (RDP) (Mironov, 2017), together with subsampling analyses and composition rules (e.g., (Wang et al., 2019)). A recurring challenge is that DP-SGD introduces DP-specific hyperparameters (notably the clipping threshold), and model utility can be highly sensitive to these choices, making tuning costly in practice (Bu et al., 2023; Wei et al., 2025).

**Methods relying on additional private queries.** A line of adaptive clipping methods, such as Adap-Clip, updates $C_t$ online using privately estimated clipping statistics, often by tracking a target quantile of gradient or update norms (Andrew et al., 2021). While this reduces manual tuning, such approaches typically require *additional private measurements*, incurring *extra privacy cost* and thus *stronger noise* to maintain the same overall privacy budget; moreover, fixed quantile targets

may become suboptimal as training dynamics evolve. Related approaches privately estimate the gradient norm distribution (e.g., via DP histograms) and select $C_t$ by optimizing an explicit bias–variance objective; DC-SGD proposes percentile- and expected-error-based variants (DC-SGD-P / DC-SGD-E), at the cost of allocating additional privacy budget to distribution estimation and introducing extra controller hyperparameters (Wei et al., 2025).

**Methods without additional queries but with added optimization components.** An orthogonal line avoids clipping-fraction estimation but introduces *additional optimization components*. AutoClip replaces clipping by transforming gradients into normalization-style updates and in specific cases rely on pretraining (Bu et al., 2023). GeoClip (Gilani et al., 2025) modifies vanilla DP-SGD by performing geometry-aware clipping in a learned transformed space, introducing extra hyperparameters and with validation limited to specific settings, the learned transformation may also require dataset-dependent adaptation to transfer across tasks. While these approaches reduce direct tuning of a clipping norm, they can shift sensitivity to other design choices (e.g., learning-rate schedules, optimizer dynamics, or stabilization heuristics), so the overall tuning burden may persist. Relatedly, methods such as DP-PSAC modify per-sample weighting/clipping rules to reduce deviation from the true batch gradient and provide convergence analysis (Xia et al., 2023).

**Clipping strategies and efficiency.** Several works study alternative clipping granularities (e.g., per-layer clipping) motivated by efficiency and scaling considerations. In particular, adaptive per-layer thresholds can match or outperform Vanilla-Clip under fixed training configurations (McMahan et al., 2018), and the effective update magnitude can become sensitive to how these per-layer thresholds (or their relative clipping ratios) are chosen, which may require careful calibration across models, optimizers, and training regimes. Related systems work accelerates per-example gradient clipping to reduce DP training overhead (Lee & Kifer, 2021); nevertheless, such acceleration primarily improves runtime and does not directly address the statistical question of selecting (potentially time-varying) clipping thresholds under a fixed privacy budget.

**Clipping bias and norm distribution shift.** Clipping introduces bias that interacts with the evolving geometry of gradients. A geometric view highlights that the gradient norm distribution can drift substantially during training and that clipping can obstruct convergence in worst cases (Chen et al., 2020). Complementary work further shows that gradient clipping can degrade utility even in the absence of injected DP noise and can increase the effective sampling noise of stochastic gradients; moreover, per-sample gradient norms may become polarized over training, with many samples having very small norms while a minority remain large (Xiao et al., 2023). Recent theory also indicates that time-varying clipping strategies can be meaningful for DP-SGD convergence (Shulgin & Richtárik, 2024).

Table 4 reports a comparison of clipping strategies between the baselines and SlaClip.

# F. Additional Experimental Details

This appendix provides additional controlled diagnostics and ablations that complement the fairly tuned comparison in Table 1. All results in this section are diagnostic: they are intended to isolate the behavior of clipping and threshold adaptation rules under matched settings, rather than to replace the main validation selected comparison.

First, Table 5 reports a controlled fixed-recipe comparison in which all methods use the default DP-SGD recipe specified in Appendix A. Unlike the main comparison, no method-specific configuration is selected from the validation grid; this makes the results useful for comparing the behavior of the clipping rules under the same optimizer, schedule, batch size, initial clipping threshold, and privacy accountant. Figure 7 complements this table by showing the corresponding training trajectories, including both test accuracy and the clipping threshold $C_t$ over training.

Second, Table 7 examines the early stage behavior of *SlaClip* on F-MNIST before the accumulated privacy loss reaches $\varepsilon = 1$ during a run targeting final privacy budget $\varepsilon = 3$. This regime contains very few DP-SGD updates: the accountant reaches $\varepsilon = 0.9975$ after only 14 training steps. At these earliest checkpoints, the released slack signal can be noisy and the Slack Indicator has limited time to stabilize; as more updates become available, *SlaClip* becomes competitive and eventually stronger, consistent with the view that the slack based controller benefits from observing sufficient gradient norm dynamics.

Finally, Tables 6, 8, 9 and 10 report controlled ablations on the main design choices of *SlaClip*. Table 6 studies the slack dimension $K$, while Table 8 further examines the interaction between practical batch sizes and admissible choices of $K$ under the design rule in Eq. (14). Table 9 reports the full sensitivity study over the initial clipping threshold $C_0$. Table 10 studies the threshold adaptation step size $\eta$. Together, these ablations characterize the robustness of *SlaClip* with respect to $\eta$, $K$, batch size, and $C_0$.

*Table 5.* **Controlled fixed-recipe comparison under shared DP-SGD settings.** Test accuracy (%) is reported as mean $\pm$ std over seeds $\{42, 43, 44\}$. Unlike the main fairly tuned comparison in Table 1, all methods here use the same manually fixed DP-SGD recipe for each dataset. We report three representative privacy budgets $\varepsilon$ under the RDP accountant, with $\delta$ fixed per dataset. Because AutoClip enforces a fixed initialization $C_0 = 1$ in our implementation, we standardize this choice across all methods for comparability. For *SlaClip*, we set the adaptation step size to $\eta = 0.5$ in this controlled fixed-recipe comparison. Within each dataset and $\varepsilon$, the best result is in **bold** and the second-best is underlined. *Model IDs:* CNN-4 = CNN-4 with AvgPool+GAP; CNN-2 = LeNet-style CNN (Conv–Pool×2 + MLP head); MLP = avg-embedding + MLP; CRNN = char-level (DP)LSTM/GRU. The default fixed recipe is specified in Appendix A.

| DATASET | MODEL | $\varepsilon$ | VANILLA-CLIP | ADAP-CLIP | DC-SGD-E | AUTOCLIP | SLACLIP-Q | SLACLIP |
|---|---|---|---|---|---|---|---|---|
| CIFAR-10 | CNN-4 | 5 | 38.31±0.64 | 57.36±1.57 | 52.43±0.33 | 38.30±0.62 | 58.00±0.51 | **59.25±1.29** |
| | | 7 | 41.50±0.24 | 63.77±0.55 | 58.54±1.18 | 41.32±0.13 | 64.01±0.96 | **65.35±1.09** |
| | | 9 | 42.41±0.12 | 67.65±0.64 | 64.38±0.58 | 42.14±0.14 | 67.84±0.79 | **68.76±0.71** |
| MNIST | CNN-2 | 1 | 61.96±1.66 | 60.25±1.85 | **73.36±1.02** | 61.98±2.71 | 57.39±4.65 | 64.37±4.54 |
| | | 2 | 95.01±0.18 | 93.41±0.46 | 94.62±0.17 | 94.57±0.12 | 92.98±0.52 | **96.21±0.35** |
| | | 3 | 96.29±0.07 | 93.36±0.37 | 94.76±0.25 | 95.97±0.09 | 93.07±0.61 | **96.84±0.25** |
| F-MNIST | CNN-2 | 1 | 54.55±0.39 | 50.19±0.91 | **60.22±1.89** | 54.56±0.60 | 53.86±2.36 | 59.86±3.26 |
| | | 2 | 83.10±0.47 | 81.26±0.15 | 82.91±0.76 | 82.32±0.58 | 80.94±0.18 | **86.14±0.22** |
| | | 3 | 84.68±0.49 | 81.25±0.08 | 86.30±0.48 | 83.94±0.46 | 81.23±0.06 | **86.88±0.09** |
| IMDB | MLP | 2 | 60.18±1.14 | **62.01±1.26** | 51.92±0.25 | 60.33±0.96 | 60.81±2.60 | 61.39±2.06 |
| | | 4 | 71.12±1.00 | 76.86±0.69 | 52.60±0.32 | 71.26±0.42 | 76.46±0.88 | **77.02±0.65** |
| | | 6 | 74.08±0.52 | 78.96±0.40 | 53.96±1.27 | 73.51±0.30 | **79.52±0.24** | 79.30±0.23 |
| NAMES | CRNN | 2 | 54.94±0.49 | 61.14±1.33 | 60.33±1.24 | 55.00±0.70 | **61.33±1.53** | 61.13±1.37 |
| | | 4 | 67.25±0.95 | 72.62±0.72 | 72.45±0.58 | 66.27±0.93 | 72.80±0.94 | **73.01±0.79** |
| | | 5 | 68.23±0.80 | 73.33±0.89 | 73.30±0.83 | 67.00±0.88 | 73.50±0.56 | **73.88±0.63** |

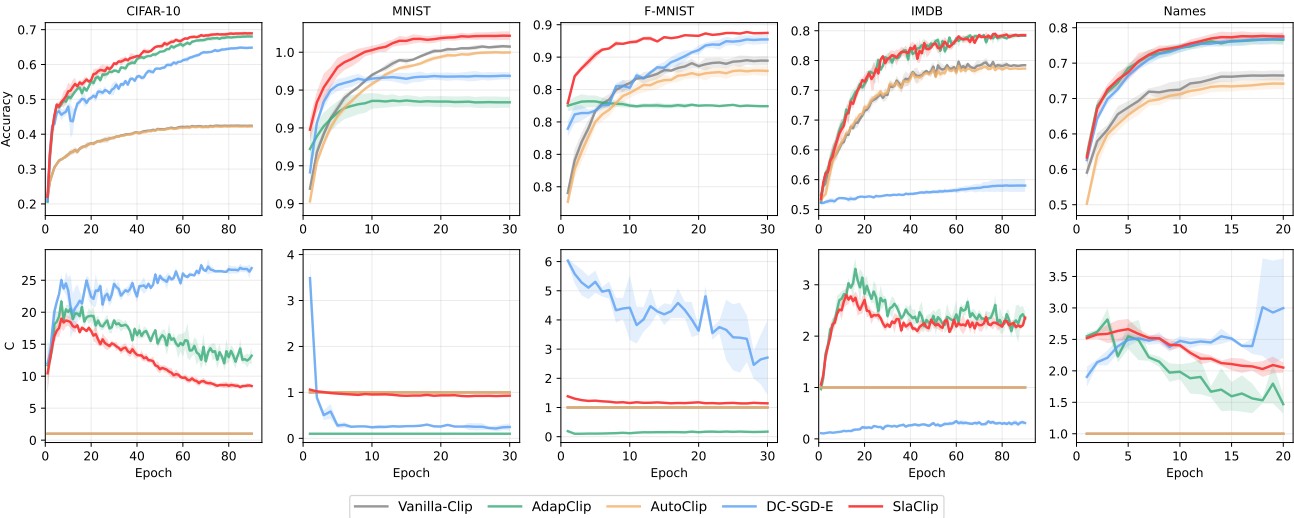

*Figure 7.* Training trajectories recorded per epoch (starting from epoch 1, last step) under matched DP-SGD settings across the five benchmarks in Table 5. Training proceeds until the maximum privacy budget for each model-dataset pair is exhausted. The top row shows test accuracy as training proceeds, and the bottom row shows the corresponding clipping threshold $C_t$. Each panel plots the mean over three random seeds, and the shaded region indicates $\pm$ std seeds. Vanilla-Clip and AutoClip keep the same clipping threshold $C_0 = 1$ fixed at one by design, while the other methods adapt $C_t$ over training.

*Table 6.* **Ablation on the slack dimension** $K$**.** Test accuracy in percent reported as mean ± std over three seeds. For each dataset and privacy budget, we sweep $K$ while keeping all other settings fixed. Within each dataset and $\varepsilon$, the best accuracy across different $K$ values is in **bold** and the second best is underlined. The final column reports the mean ± std across the set of $K$ values in the row, computed over the $K$-wise mean accuracies.

| DATASET | $\varepsilon$ | $K=5$ | $K=10$ | $K=20$ | $K=30$ | $K=40$ | $K=60$ | $K=100$ | MEAN+STD |
|---|---|---|---|---|---|---|---|---|---|
| **CIFAR10** | 5 | 58.06±2.25 | 59.07±2.03 | 58.99±1.65 | **59.25±1.29** | 58.75±1.17 | 58.87±0.73 | 58.84±1.84 | 58.83±0.38 |
| | 7 | 64.16±1.03 | 65.25±2.56 | 65.53±1.73 | 65.35±1.09 | 65.62±1.42 | **65.92±1.15** | 65.60±0.83 | 65.35±0.57 |
| | 9 | 68.09±0.64 | **69.07±0.92** | 68.99±0.57 | 68.76±0.71 | 68.69±1.39 | 69.03±0.64 | 68.55±0.39 | 68.74±0.35 |
| **MNIST** | 1 | **64.88±4.74** | 63.93±3.81 | 64.37±4.54 | 63.18±5.02 | 62.64±5.02 | 61.83±4.50 | 61.71±5.85 | 63.22±1.23 |
| | 2 | 96.31±0.25 | **96.31±0.34** | 96.21±0.35 | 96.01±0.22 | 96.01±0.24 | 95.85±0.38 | 95.73±0.47 | 96.06±0.22 |
| | 3 | **97.00±0.23** | 96.91±0.25 | 96.84±0.25 | 96.65±0.24 | 96.65±0.14 | 96.50±0.30 | 96.29±0.37 | 96.69±0.25 |
| **IMDB** | 2 | 61.12±2.21 | 61.39±2.06 | 61.47±1.96 | 61.47±1.93 | **61.64±1.92** | 61.52±2.01 | 61.46±1.91 | 61.44±0.16 |
| | 4 | 76.57±0.82 | **77.02±0.65** | 76.75±0.69 | 76.63±0.72 | 76.57±0.70 | 76.56±0.72 | 76.61±0.76 | 76.67±0.17 |
| | 6 | **79.56±0.27** | 79.30±0.23 | 79.06±0.30 | 78.98±0.31 | 78.92±0.32 | 78.90±0.33 | 78.89±0.27 | 79.09±0.25 |

*Table 7.* Behavior under sub-1 privacy budgets on F-MNIST. Each row reports the test accuracy (%) after a given number of DP-SGD steps, where one step denotes one training iteration on one minibatch. Results are reported as mean ± std over three random seeds. The table illustrates the early stage regime where the privacy budget is close to $\varepsilon = 1$ after only a small number of updates.

| Step | $\varepsilon$ | Vanilla-Clip | DC-SGD-E | *SlaClip* |
|---|---|---|---|---|
| 1 | 0.9160 | 16.79 ± 2.98 | 16.79 ± 2.99 | 16.74 ± 2.92 |
| 2 | 0.9341 | 20.62 ± 4.71 | 23.33 ± 3.39 | 21.39 ± 4.16 |
| 3 | 0.9455 | 24.37 ± 3.40 | 31.03 ± 4.51 | 28.26 ± 2.73 |
| 4 | 0.9545 | 29.26 ± 4.20 | 36.66 ± 7.23 | 33.08 ± 5.59 |
| 5 | 0.9608 | 32.46 ± 6.30 | 37.50 ± 3.42 | 39.54 ± 4.10 |
| 6 | 0.9670 | 36.16 ± 7.32 | 42.67 ± 2.53 | 45.85 ± 6.00 |
| 7 | 0.9719 | 41.16 ± 6.94 | 47.81 ± 5.61 | 49.93 ± 7.76 |
| 8 | 0.9763 | 45.18 ± 6.17 | 50.49 ± 7.30 | 47.27 ± 4.58 |
| 9 | 0.9807 | 49.17 ± 7.24 | 50.11 ± 4.95 | 49.61 ± 3.55 |
| 10 | 0.9849 | 51.19 ± 7.70 | 55.13 ± 5.50 | 57.06 ± 1.88 |
| 11 | 0.9881 | 51.36 ± 7.74 | 55.06 ± 5.53 | 56.34 ± 1.55 |
| 12 | 0.9912 | 51.22 ± 8.00 | 54.53 ± 2.36 | 58.43 ± 4.56 |
| 13 | 0.9944 | 51.70 ± 7.78 | 54.41 ± 3.14 | 60.31 ± 1.93 |
| 14 | 0.9975 | 52.68 ± 8.62 | 54.80 ± 2.56 | 60.26 ± 2.97 |

*Table 8.* **Practical batch size and admissible-$K$ study.** For each batch size, we report results at three privacy levels; the shown $\varepsilon$ is the *achieved* privacy budget under the RDP accountant. Since varying the batch size also changes the sampling rate $q = B/|D|$, and hence the incurred privacy cost, we adjust the privacy constraint accordingly to ensure a fair comparison under each experimental setting.

| Method | $B=512$ ($K=20$) | | $B=1024$ ($K=30$) | | $B=2048$ ($K=50$) | |
|---|---|---|---|---|---|---|
| | Acc (%) | $\varepsilon$ | Acc (%) | $\varepsilon$ | Acc (%) | $\varepsilon$ |
| Vanilla-Clip | **44.36±0.41** | 4 | 38.31±0.64 | 5 | 33.83±0.31 | 6 |
| | 44.94±0.39 | 5 | 41.50±0.24 | 7 | 35.35±1.24 | 8 |
| | 45.69±0.23 | 6 | 42.41±0.12 | 9 | 36.93±0.80 | 10 |
| Adap-Clip | 41.25±5.42 | 4 | 57.36±1.57 | 5 | **57.13±2.06** | 6 |
| | 45.66±4.67 | 5 | 63.77±0.55 | 7 | 63.23±0.10 | 8 |
| | 51.86±5.22 | 6 | 67.65±0.64 | 9 | 67.11±0.18 | 10 |
| AutoClip | 44.18±0.51 | 4 | 38.30±0.62 | 5 | 33.80±0.35 | 6 |
| | 44.71±0.31 | 5 | 41.32±0.13 | 7 | 35.34±1.22 | 8 |
| | 45.45±0.17 | 6 | 42.14±0.14 | 9 | 36.86±0.78 | 10 |
| DC-SGD-E | 42.55±1.03 | 4 | 52.43±0.33 | 5 | 54.22±1.73 | 6 |
| | 48.93±0.95 | 5 | 58.54±1.18 | 7 | 60.43±1.11 | 8 |
| | 54.91±0.50 | 6 | 64.38±0.58 | 9 | 64.58±0.71 | 10 |
| SLACLIP | 43.92±2.58 | 4 | **59.25±1.29** | 5 | 56.67±1.17 | 6 |
| | **50.69±2.51** | 5 | **65.35±1.09** | 7 | **63.76±0.63** | 8 |
| | **55.70±2.48** | 6 | **68.76±0.71** | 9 | **67.46±0.73** | 10 |

*Table 9.* **(full results): accuracy under different initial thresholds** $C_0$**.** Test accuracy (mean±std, in %) over three seeds (42/43/44) when sweeping the initialization $C_0 \in \{0.1, 1, 5, 10, 20\}$. We report three privacy budgets per dataset under the RDP accountant: CIFAR-10 ($\varepsilon$=3/5/9), MNIST (1.1/1.5/3), F-MNIST (F-MNIST) (1.1/1.5/3), and IMDB (2/4/6). Within each $(C_0, \varepsilon)$ setting, the best method is in **bold** and the second best is underlined. AutoClip is omitted since it uses a fixed $C_0$=1 in our implementation and is not comparable in the sweep. For *SlaClip*, this sweep is conducted with the Adap-Clip recommended step size $\eta$=0.2, which differs from the controlled fixed-recipe comparison in Table 5; SlaClip's stability w.r.t. $\eta$ is studied in Table 10.

| $C_0$ | Method | CIFAR-10 ($\varepsilon$=3/5/9) | MNIST ($\varepsilon$=1.1/1.5/3) | F-MNIST ($\varepsilon$=1.1/1.5/3) | IMDB ($\varepsilon$=2/4/6) |
|---|---|---|---|---|---|
| 0.1 | Vanilla-Clip | 23.47±1.07 / 26.04±0.37 / 26.95±0.43 | 62.36±3.20 / 82.80±0.19 / 88.09±0.22 | 55.75±5.61 / 68.49±0.90 / 73.08±0.31 | 51.29±0.48 / 51.14±0.29 / 51.15±0.31 |
| | Adap-Clip | 48.26±3.07 / 56.70±1.92 / 67.67±0.46 | 87.98±0.57 / 90.01±0.24 / 90.22±0.02 | 74.74±1.47 / 81.46±0.17 / 80.98±0.20 | **61.43**±**1.88** / 75.86±1.53 / 79.12±0.37 |
| | DC-SGD-E | 44.73±3.24 / 52.40±1.02 / 63.63±0.57 | **88.50**±**0.15** / **94.16**±**0.15** / 94.42±0.53 | **77.46**±**1.60** / 80.75±0.41 / **86.25**±**0.12** | 52.03±0.15 / 52.64±0.27 / 54.59±2.61 |
| | **SlaClip** | **51.20**±**1.78** / **58.73**±**0.55** / **69.20**±**0.46** | 87.57±0.81 / 93.41±0.09 / **96.15**±**0.16** | 73.69±0.61 / **83.56**±**0.15** / 85.89±0.13 | 61.34±2.07 / **76.93**±**0.63** / **79.31**±**0.24** |
| 1 | Vanilla-Clip | 33.79±1.02 / 38.31±0.64 / 42.41±0.12 | 86.72±0.33 / 92.93±0.23 / **96.29**±**0.07** | 71.96±0.52 / 80.27±1.13 / 84.68±0.49 | 60.18±1.14 / 71.12±1.00 / 74.08±0.52 |
| | Adap-Clip | 49.46±1.39 / 57.36±1.57 / 67.65±0.64 | **89.44**±**0.55** / 90.59±0.43 / 93.36±0.37 | 77.85±0.77 / 81.59±0.25 / 81.25±0.08 | **62.01**±**1.26** / 76.86±0.69 / 78.96±0.40 |
| | DC-SGD-E | 46.09±2.21 / 52.43±0.33 / 64.38±0.58 | 88.88±0.40 / **94.24**±**0.32** / 94.76±0.25 | **78.01**±**1.60** / 81.08±0.57 / **86.30**±**0.48** | 51.92±0.25 / 52.60±0.32 / 53.96±1.27 |
| | **SlaClip** | **51.54**±**1.40** / **58.69**±**1.88** / **69.00**±**0.61** | 89.02±0.33 / 93.59±0.11 / 96.10±0.11 | 76.03±2.00 / **84.15**±**0.29** / 86.00±0.14 | 61.50±2.01 / **76.72**±**0.75** / **79.07**±**0.32** |
| 5 | Vanilla-Clip | 47.65±0.52 / 56.35±0.51 / 63.89±0.56 | 88.49±1.15 / 91.44±0.50 / 96.26±0.15 | 77.07±1.29 / 81.05±1.36 / 86.46±0.16 | 61.56±1.99 / 73.09±1.38 / 78.26±0.45 |
| | Adap-Clip | 48.94±2.18 / 55.78±0.21 / 67.24±0.60 | **91.37**±**0.93** / 92.83±0.82 / 92.54±0.55 | **79.31**±**0.69** / 82.39±0.21 / 81.58±0.24 | **61.60**±**1.86** / 75.69±1.51 / 79.13±0.38 |
| | DC-SGD-E | 46.43±2.05 / 52.79±0.78 / 63.51±1.28 | 88.67±1.37 / 94.33±0.39 / 94.70±0.18 | 78.85±1.02 / 80.67±0.49 / 86.49±0.48 | 52.03±0.11 / 52.62±0.25 / 54.20±1.95 |
| | **SlaClip** | **51.89**±**2.05** / **58.62**±**0.59** / **69.39**±**0.35** | 90.06±0.78 / **94.68**±**0.25** / **96.55**±**0.19** | 79.30±0.34 / **84.90**±**0.10** / **86.59**±**0.04** | 61.30±2.17 / **76.94**±**0.62** / **79.32**±**0.23** |
| 10 | Vanilla-Clip | **53.83**±**1.11** / **61.68**±**1.23** / **69.41**±**0.13** | 86.56±0.71 / 87.64±1.29 / 94.34±0.09 | 77.69±1.63 / 76.98±0.32 / 84.33±0.29 | 58.39±2.61 / 64.25±0.37 / 70.45±0.75 |
| | Adap-Clip | 48.77±1.35 / 56.44±1.60 / 68.29±0.40 | **91.26**±**1.07** / 92.80±0.98 / 92.59±0.94 | 79.21±1.03 / 82.34±0.59 / 81.54±0.23 | **61.94**±**1.98** / 75.60±1.49 / 79.11±0.36 |
| | DC-SGD-E | 46.27±1.62 / 53.30±1.14 / 64.00±0.25 | 89.73±0.97 / 94.50±0.31 / 94.50±0.64 | 78.30±1.79 / 80.59±1.09 / 86.51±0.22 | 52.06±0.05 / 52.64±0.27 / 54.10±1.72 |
| | **SlaClip** | 51.36±0.85 / 59.02±0.74 / 69.38±0.40 | 90.89±0.98 / **95.06**±**0.28** / **96.79**±**0.21** | 79.90±0.46 / **85.04**±**0.26** / **86.74**±**0.12** | 61.67±1.97 / **76.97**±**0.63** / **79.31**±**0.27** |
| 20 | Vanilla-Clip | 50.52±0.30 / 52.74±2.13 / 66.46±0.58 | 83.28±2.17 / 80.03±2.03 / 86.97±6.70 | 68.15±1.36 / 39.71±15.72 / 36.22±5.81 | 52.65±2.10 / 51.53±1.98 / 55.91±2.73 |
| | Adap-Clip | 50.03±0.43 / 56.65±0.93 / 68.37±0.31 | 91.78±0.35 / 93.62±0.49 / 94.08±0.71 | 78.90±1.60 / 82.38±0.62 / 81.62±0.27 | 61.68±2.74 / 75.52±1.59 / 79.13±0.33 |
| | DC-SGD-E | 47.32±1.49 / 50.72±2.20 / 63.52±0.45 | 90.20±1.00 / 94.44±0.38 / 94.30±0.38 | 78.09±2.20 / 81.66±2.66 / 86.76±0.50 | 52.08±0.12 / 52.61±0.27 / 53.75±1.12 |
| | **SlaClip** | **51.85**±**2.22** / **58.11**±**0.73** / **68.87**±**0.53** | **91.89**±**0.36** / **95.69**±**0.10** / **97.29**±**0.05** | **79.93**±**0.07** / **85.16**±**0.18** / **86.96**±**0.17** | **62.14**±**1.98** / **77.04**±**0.69** / **79.27**±**0.34** |

*Table 10.* **Ablation on the threshold adaptation step size** $\eta$ **in** *SlaClip***.** Test accuracy in percent reported as mean ± std deviation over three seeds. We sweep $\eta \in \{0.05, 0.1, 0.2, 0.5, 1.0\}$ while keeping all other training and privacy settings matched. For each dataset we evaluate three representative privacy budgets under the RDP accountant, matching the budgets used in Table 5. Within each dataset and $\varepsilon$, the best accuracy across $\eta$ values is in **bold**. The final column reports the mean ± std across the set of $\eta$ values in the row, computed over the $\eta$-wise mean accuracies.

| DATASET | $\varepsilon$ | $\eta = 0.05$ | $\eta = 0.1$ | $\eta = 0.2$ | $\eta = 0.5$ | $\eta = 1$ | MEAN+STD |
|---|---|---|---|---|---|---|---|
| CIFAR10 | 5 | 58.71±1.33 | 58.42±0.87 | 58.69±1.88 | **59.25**±**1.29** | 59.22±1.25 | 58.78±0.29 |
| | 7 | 65.45±1.11 | 65.29±1.30 | **65.89**±**0.89** | 65.35±1.09 | 65.66±1.53 | 65.63±0.26 |
| | 9 | **69.33**±**0.81** | 68.73±0.42 | 69.00±0.61 | 68.76±0.71 | 68.94±0.83 | 69.05±0.24 |
| MNIST | 1 | 63.13±4.01 | 64.15±2.74 | **64.37**±**3.58** | 64.37±4.54 | 54.17±8.24 | 61.80±4.30 |
| | 2 | 94.87±0.29 | 94.95±0.22 | 95.09±0.24 | 96.21±0.35 | **96.52**±**0.03** | 95.49±0.74 |
| | 3 | 96.00±0.09 | 96.01±0.06 | 96.10±0.11 | 96.84±0.25 | **97.17**±**0.08** | 96.39±0.52 |
| F-MNIST | 1 | 54.20±9.32 | 54.16±7.14 | 55.71±3.22 | **59.86**±**3.26** | 49.85±6.18 | 54.94±3.94 |
| | 2 | 84.65±0.72 | 84.87±0.65 | 85.11±0.70 | **86.14**±**0.22** | 85.75±0.82 | 85.25±0.54 |
| | 3 | 85.67±0.25 | 85.85±0.29 | 86.00±0.14 | **86.88**±**0.09** | 86.49±0.31 | 86.16±0.46 |
| IMDB | 2 | 61.15±2.29 | 61.35±2.12 | 61.50±2.01 | 61.39±2.06 | **61.51**±**1.88** | 61.39±0.15 |
| | 4 | 76.69±0.74 | 76.71±0.74 | 76.72±0.75 | **77.02**±**0.65** | 76.89±0.51 | 76.75±0.08 |
| | 6 | 79.10±0.32 | 79.08±0.33 | 79.07±0.32 | **79.30**±**0.23** | 79.07±0.33 | 79.08±0.02 |

