# OpenReview forum: "SlaClip: Gradient Norm Slacks can be Indicator for Adaptive Clipping in DP-SGD"
_ICML.cc/2026/Conference — ICML 2026 spotlight_

### Official Review · Reviewer_ATRi · 2026-03-03

**Soundness:** 3
**Presentation:** 2
**Significance:** 4
**Originality:** 4
**Overall Recommendation:** 5
**Confidence:** 4

**Summary:**

The paper considers the problem of adaptively choosing the clipping norm in DP-SGD. The authors identify a weakness of past works on adaptive clipping is that they involve additional queries which require an increased noise multiplier to achieve the same per-round privacy. The authors propose SlaClip, a method for adaptive clipping which encodes additional information into gradient queries and hence doesn't require additional noise. SlaClip concatenates to the clipped gradient g a vector s which encodes information about the slack in clipping, such that (g, s) has norm at most C still (e.g. if g is clipped, s will be all zeros). s is split into multiple coordinates so that even after averaging across multiple clipped gradients and noised, s still encodes the following information (i) what fraction of gradients were not clipped and (ii) what fraction of gradients have very small norms (and hence should not have much impact on clipping). The authors include a suggested hyperparameter choice for their method to avoid the need for additional hyperparameter tuning. The authors use (ii) to estimate a percentile of norms to use as a clipping threshold, and then use an update rule to get closer to this percentile based on (i). The paper concludes with experiments comparing SlaClip to other adaptive clipping methods and vanilla DP-SGD. SlaClip achieves higher accuracies, demonstrates better robustness to the initial clipping threshold, and also robustness to different batch sizes.

**Compliance With Llm Reviewing Policy:**

Affirmed.

**Final Justification:**

I am already in support of accepting the paper and the rebuttal process has not raised any new concerns for me, and made aspects of the algorithm design clearer. I remain in support of acceptance.

**Key Questions For Authors:**

* It seems like Equation 8 could be tightened giving better utility for free: Instead of having $max(C - \|g\|, 0) = a \lambda + b$, why not have it be something like $max(C^2 - \|g\|^2) = a \lambda^2 + b^2$? This way you would always have a norm C vector, i.e. you'd be maximizing the signal-to-noise ratio on the auxiliary coordinates.
* Only two of the auxiliary coordinates are used in SlaClip's definition. Why not present SlaClip restricted to these two coordinates? This also seems like you would get higher utility for free as you could increase the value of these two coordinates for a given gradient while still satisfying the same sensitivity bound.

**Limitations:**

Yes

**Strengths And Weaknesses:**

Strengths:
* Cleanly identifies a key weakness of past work, the added privacy cost of additional queries, and gives an elegant approach that cleanly subverts that weakness.
* Beyond just the realization that gradients which are not clipped can have additional coordinates added to them at no cost, the fact that multiple coordinates can be used to encode information about the distribution of unclipped gradient norms and the changes to the update rule based on this information are meaningful technical contributions.
* Paper contributes a well-motivate
* Experiments demonstrate the advantage over other methods
* Experiments ablate nicely, demonstrating robustness to various actors

Weaknesses:
* SlaClip is written in a general form where it stores K coordinates of information, but only 2 are used in the final algorithm. This seems like it sacrifices utility unnecessary, and also makes the presentation a bit confusing. See "Key Questions For Authors" for more details.
* I would suggest not anchoring Theorem 3.1 to RDP accounting, since the statement holds for other tighter forms of DP/accounting such as f-DP and PLD accounting. I think this doesn't even need to be a theorem, just an observation that any bounded-sensitivity Gaussian mechanisms with the same noise multipliers satisfy the same DP guarantees.

---

> ### Author Rebuttal · Authors · 2026-03-31
>
> Thank you for the supportive and insightful review.
>
> **Main clarification (W1, Q2).** We appreciate the reviewer's careful reading and the insightful presentation suggestions. We will revise the exposition accordingly and make the highlighted points clearer. For the two-coordinate question, (1) the current controller uses two coordinates because they are the two most directly actionable signals for threshold updates: $\hat{s_{t,1}}$ estimates the CDF mass nearest the current threshold and indicates whether $C_t$ is too small or too large, while $s_{t,K}/C_t$ estimates near-zero mass and indicates how much of the batch is likely to be noise-dominated; (2) we still retain the remaining coordinates because they preserve a useful discretized CDF estimate without increasing the privacy cost under the same accountant, which is useful for validating the released CDF structure now (as monitored in Fig. 5) and may support better clipping-threshold update rules in future work. The additional coordinates are therefore not redundant dimensions; they enrich the released signal while preserving the same privacy cost under the same accountant, and the runtime overhead remains modest (see the **CIFAR-10 Runtime Comparison** reported to Reviewer uJLz).
>
> **Why keep the other coordinates?** The broader $K$-coordinate construction is not only a presentation choice; it is the discretized CDF estimator itself. In the paper, each coordinate corresponds to an equal-width CDF bin, so the interior coordinates are part of the released CDF profile rather than redundant extras. This matters for two reasons. First, the *Choosing $K$* rule relies on the multi-coordinate CDF structure: the monotonicity of neighboring coordinates is exactly what motivates the Eq. 14 upper bound that prevents noise from degrading the released CDF. Second, once interior bins are used in the controller, the update rule must trade off higher CDF resolution against stronger Gaussian-noise amplification, since larger $K$ improves resolution but can also degrade the quality of the released CDF. Our current controller therefore uses the two most directly actionable coordinates while keeping the full Slack Indicator structure intact. This preserves the clean adaptive-clipping plug-in design and leaves richer multi-bin controllers to future work.
>
> **Equation 8 (Q1).** Thank you for this insightful suggestion. Tightening Eq. 8 so that the auxiliary signal always lies on the norm-$C$ boundary is technically appealing, but it changes the structure of the binned CDF: the bins would no longer be equal-width in gradient norm, so the indicator would correspond to uneven bin sizes. Under noise, this makes the underlying distribution harder to model and also makes the non-decreasing CDF property harder to guarantee. That, in turn, makes the choice of $K$ more difficult. In other words, the apparent *free utility* gain comes with a more difficult estimation problem. We therefore view this as a strong direction for future work rather than a clear strict improvement to the current design.
>
> **Theorem 3.1 (W2).** We agree that Theorem 3.1 is better framed as an observation about bounded-sensitivity Gaussian releases, not as an RDP-specific theorem. The same intuition extends beyond the particular accountant used in the main text. We will rephrase it accordingly.

---

> > ### Author Rebuttal · Reviewer_ATRi · 2026-04-03
> >
> > After trying to write up a more formal version of my suggestion I see what breaks down when you try to add more signal to the two coordinates you are using. Effectively, you want to compute something like the median so the signal of an example that has a lot of slack and only has $\lambda$ slack should still be the same in the first coordinate. Thanks to the authors for clarifying this. I am happy to retain my score.

---

> > > ### Author Response · Authors · 2026-04-05
> > >
> > > Thank you for the thoughtful follow-up and positive assessment. We especially appreciate the time you spent working through the formulas-we truly enjoyed the discussion.
> > >
> > > We also agree that SlaClip is appealing because it is a plug-and-play mechanism and provides adaptivity without additional privacy cost.
> > >
> > > Thank you again for helping improve the paper.

---

### Official Review · Reviewer_uJLz · 2026-03-12

**Soundness:** 4
**Presentation:** 4
**Significance:** 2
**Originality:** 3
**Overall Recommendation:** 5
**Confidence:** 3

**Summary:**

The paper introduces a new adaptive clipping method called SlaClip. It works based on the concept of slack that they introduce, which is the difference between the clipping threshold C_t and the gradient norm. This slack information is encoded into a K-dimensional slack vector
which is then appended to the per-sample gradient. This extended gradient is proven to preserve the same ℓ_2 sensitivity as vanilla DP-SGD, hence the Gaussian mechanism's privacy cost bound is unchanged. In summary, the main contribution in the paper is that, unlike the prior work, the proposed SlaClip can achieve adaptive clipping without introducing additional queries or gradient transformations by leveraging slack information. The authors also provide extensive empirical analysis and ablation showing the effectiveness of the proposed algorithm.

**Compliance With Llm Reviewing Policy:**

Affirmed.

**Ethical Review Concerns:**

yes

**Final Justification:**

My questions have been answered, and given the paper's innovative line of thinking, I have increased my score to accept.

**Key Questions For Authors:**

Q1. Can you evaluate at ϵ < 1 (e.g., ϵ = 0.5)? Given the observed weakness at ϵ = 1, understanding behavior in the strong-privacy regime is insightful for the paper's practicality. If its infeasible during rebuttal then explicitly addressing the low-epsilon range where SlaClip is underperforming and acknowledging this in discussion/limitation would partially address the concern.

Q2. In the Appendix Table 5, ablation shows that C_0 = 10 helps SlaClip under tight privacy budgets, but the Table 1 in main paper uses
C_0 = 1 where it underperforms in some setting. Is the low-epsilon weakness an artifact of initialization? If so, this raises a follow-up question about how should a practitioner select C_0 without a hyperparameter sweep in large/complex models?

Q3. You claim SlaClip is a drop-in replacement but the paper reports no wall-clock time or memory comparisons. Appending dimensions to every per-sample gradient and sampling noise in R^{d+K} is not free as Opacus already introduces significant overhead for per-sample gradient computation. Can you report (or discuss) training time and memory overhead? This would strengthen the practical contribution claim

**Limitations:**

yes

**Strengths And Weaknesses:**

Strengths

S1. The core idea of encoding slack information into extra gradient dimensions and releasing it via the same Gaussian mechanism is genuinely novel. The theoretical foundation is clean and easy to follow. Most importantly, the proposed method is a drop-in replacement mechanism for vanilla DP-SGD, which is what makes the paper strong and practical.

S2. Experiments and ablation studies are insightful and thorough.

S3. The paper is easy to read and follow.

Major Weaknesses

W1. The empirical evaluations is limited to small models and datasets. For ICML, the absence of relatively modern architectures (e.g., ViT) raises concerns about the practicality of the contribution in a real-world setting. It is not clear that whether the proposed method remains effective for large-scale models.

W2. SlaClip underperforms in the low-epsilon regime (eps=1 MNIST/F-MNIST), the authors acknowledge this in the paper adequately. However, the low-epsilon regime is actually where adaptive clipping matters most, as the privacy budget is tight. I can see that such
low-epsilon weakness is not universal (ϵ = 2 on MNIST achieves best). However, pattern across the results show that SlaClip is most consistent at moderate/high epsilon (e.g., ϵ = 5–9 on CIFAR-10,  ϵ = 4–6 on IMDB). If this is related to the C_0, then the drop-in
replacement argument is weakened as one needs to do hyperparameter tuning beforehand.

W3. Prior work has raised concerns about the practical meaningfulness of privacy guarantees in such large epsilon range. Jayaraman & Evans (USENIX, 2019) showed that settings providing acceptable utility tend to offer weak privacy protection, and Ponomareva et al. (JAIR, 2023) argues that even the relaxed convention of ϵ ≤ 10 in deep learning represents an already-weakened standard compared to the ϵ <  1. This raises question about paper's significance as the paper never evaluates below ϵ < 1. It is not clear that whether the method is still viable at sub-1 regime is unknown.

Minor Weaknesses

W4. Computational overheads are not discussed. Since the algorithm is is mentioned as a drop-in replacement, the paper would benefit from such a overhead discussion

---

> ### Author Rebuttal · Authors · 2026-03-31
>
> Thank you for the thoughtful review.
>
> **Clarification on strong privacy guarantee (W2, W3, Q1).** (1) we agree that ε<1 is the stronger formal privacy regime, but USENIX 2019 surveys the low-budget line of work built around output/objective perturbation under strong-convexity assumptions, and notes that such methods are *"only applicable to simple binary classification tasks"*; the same Section 2.3 then states that *"high privacy budgets are required for non-convex learning algorithms, such as deep learning"*. JAIR 2023 similarly labels $ε \le 1$ as *"Tier 1: Strong formal privacy guarantees"* and describes Tier 2 as *"the currently undocumented but still widely used goal for DP ML models of achieving an $ε \le 10$"* (Section 5.2.1). This is also the regime used by most DP-SGD baselines, including Adap-Clip, Auto-Clip, and DC-SGD. (2) Beyond those empirical and commonly used settings, we appreciate your interest in the stronger formal privacy regime, and therefore add ε<1 results below.
>
> **Experiments with ε<1**
>
> In our benchmark-scale experiments, the sub-1 regime is so restrictive that the privacy budget is exhausted before the model completes a full pass over the training data, as shown in the table below. Here, one step denotes a single training iteration on one batch. We observe that the privacy budget is nearly exhausted after only 14 steps, which covers only ~12% of the training set. SlaClip is not uniformly best at the earliest steps, but becomes competitive and eventually stronger later as ε approaches 1.
>
> **F-MNIST**
>
> | Step | ε | Vanilla-Clip | DC-SGD-E | SLaClip |
> |---:|---:|---:|---:|---:|
> | 1 | 0.9160 | 16.79 ± 2.98 | 16.79 ± 2.99 | 16.74 ± 2.92 |
> | 2 | 0.9341 | 20.62 ± 4.71 | 23.33 ± 3.39 | 21.39 ± 4.16 |
> | 3 | 0.9455 | 24.37 ± 3.40 | 31.03 ± 4.51 | 28.26 ± 2.73 |
> | 4 | 0.9545 | 29.26 ± 4.20 | 36.66 ± 7.23 | 33.08 ± 5.59 |
> | 5 | 0.9608 | 32.46 ± 6.30 | 37.50 ± 3.42 | 39.54 ± 4.10 |
> | 6 | 0.9670 | 36.16 ± 7.32 | 42.67 ± 2.53 | 45.85 ± 6.00 |
> | 7 | 0.9719 | 41.16 ± 6.94 | 47.81 ± 5.61 | 49.93 ± 7.76 |
> | 8 | 0.9763 | 45.18 ± 6.17 | 50.49 ± 7.30 | 47.27 ± 4.58 |
> | 9 | 0.9807 | 49.17 ± 7.24 | 50.11 ± 4.95 | 49.61 ± 3.55 |
> | 10 | 0.9849 | 51.19 ± 7.70 | 55.13 ± 5.50 | 57.06 ± 1.88 |
> | 11 | 0.9881 | 51.36 ± 7.74 | 55.06 ± 5.53 | 56.34 ± 1.55 |
> | 12 | 0.9912 | 51.22 ± 8.00 | 54.53 ± 2.36 | 58.43 ± 4.56 |
> | 13 | 0.9944 | 51.70 ± 7.78 | 54.41 ± 3.14 | 60.31 ± 1.93 |
> | 14 | 0.9975 | 52.68 ± 8.62 | 54.80 ± 2.56 | 60.26 ± 2.97 |
>
> A detailed explanation to the initial privacy budget after first batch iteration starts at 0.9160 is: under the Opacus RDP accountant, the privacy loss after the first optimizer step is already $ε_1 \approx 0.916$. In our setup, $σ=1.0$, $δ=10^{-5}$, and $q=1/118\approx 0.00847$. After one step, the accountant selects $\alpha^\star=10.3$ with $RDP_{10.3}=0.030962$, and converts it to $(ε,δ)$-DP via
> $$
> ε(\alpha)=\mathrm{RDP}_{\alpha}-\frac{\log δ+\log \alpha}{\alpha-1}+\log\left(\frac{\alpha-1}{\alpha}\right),
> $$
> which yields $ε_1\approx 0.9160$. Hence, the curve naturally starts near $0.916$.
>
> **Modern architectures. (Weakness 1)** We agree that experiments on ViT-style models would strengthen the paper. To verify that the effectiveness of SLaClip is **not limited to CNNs**, we follow *Differentially Private Sharpness-Aware Training (ICML 2023)*. Specifically, we conducted an experiment using the **ViT-style architecture** adopted in this ICML 2023 paper, and compared SlaClip with DP-SAT, an efficient training method proposed in that paper:
>
> **ViT on CIFAR-100**
>
> | Privacy Budget| Model | # Params | Vanilla-Clip | SlaClip | DP-SAT |
> |---|---|---:|---:|---:|---:|
> | ε = 2 | CrossViT-small-240 | 26.3M | 71.14 ± 0.38 | 71.94 ± 0.67 | 71.38 ± 0.23 |
>
> **Q2. Choosing $C_0$.** Appendix Table 5 already suggests the practical recommendation: under tight privacy, $C_0 = 10$ is a better default than $C_0 = 1$, so the low-ε weakness is at least partly an initialization issue.
>
> **W4 / Q3. Runtime and memory overhead.** The privacy cost is unchanged because SlaClip uses the same release and preserves the same sensitivity bound. The computational cost is not zero, but it is modest because $K \ll d$. We benchmarked three training pipelines on CIFAR-10 with batch size 1024 on A100 GPU. Each run lasted for 5 epochs, the first epoch was excluded as warm-up, and the remaining 4 epochs were used for timing. We repeated each experiment 3 times and report mean ± standard deviation. The results show that SlaClip incurs only about 10% additional computational overhead relative to vanilla DP-SGD.
>
> **CIFAR-10 Runtime Comparison**
>
> | Pipeline | Measured Epochs / Run | Repeats | Time / Epoch (s) | Mean Step Time (s) | Throughput (samples/s) |
> |---|---:|---:|---:|---:|---:|
> | nonDP | 4 | 3 | 5.957 ± 0.015 | 0.1216 ± 0.0003 | 8393.16 ± 20.70 |
> | Vanilla-Clip | 4 | 3 | 7.726 ± 0.156 | 0.1577 ± 0.0032 | 6473.59 ± 132.23 |
> | SlaClip | 4 | 3 | 8.193 ± 0.278 | 0.1672 ± 0.0057 | 6107.46 ± 203.53 |

---

> > ### Author Rebuttal · Reviewer_uJLz · 2026-04-03
> >
> > My questions have been answered.
> >
> > But I do want to insist that "Tier 2" privacy equal "No privacy." We should not lead an uninformed reader on a wrong path of interpretation. Tier 2 privacy means large eps and the reduced hypothesis testing problem from the DP framework gives the adversary a significant advantage (you would be surprised if calculating actual numbers).
> >
> > As a community we cannot hide behind misleading policies/recommendations and we need to set this straight. Yes, originally, DP-SGD showed a fantastic step forward asking for more future work.
> >
> > I encourage the authors to re-evaluate their argument and write about what eps range is really needed in practical scenarios (if we want to make theoretical guarantees and not heuristic privacy guarantees that evaluate SOTA MIA attacks).
> >
> > I look forward to hear your final thoughts on this.

---

> > > ### Author Response · Authors · 2026-04-04
> > >
> > > We are happy that the concerns have been fully resolved, and sincerely thank the reviewer for the careful follow-up.
> > >
> > > **Tier 2 Privacy**
> > >
> > > We agree that the $ε<1$ regime is worth studying. In our view, this remains an active and promising research question in the DP-SGD literature rather than a settled one. Most of the current DP-SGD literature, including the adaptive clipping baselines in our paper (Adap-Clip, Auto-Clip, DCSGD), still evaluates mainly at $ε > 1$ rather than exclusively in the sub-1 regime [1]. We also appreciate the reviewer’s point that *large $ε$ can imply a substantial adversarial advantage under the reduced hypothesis-testing interpretation of DP*. This is exactly why the practical meaning of different $ε$ regimes should be stated carefully in existing DP-SGD studies.
> > >
> > > Similar to the above work, our paper focuses on **training from scratch**, where DP-SGD in the $ε < 1$ regime often leads to models with extremely poor accuracy in practice. We therefore believe that studying the sub-1 regime is particularly meaningful under the **pretrain + DP-SGD fine-tuning** setting. A concrete example is as follows.
> > >
> > > **Sub-1 regime**
> > >
> > > A prominent example is DeepMind's paper [2], which reports 83.8\% top-1 accuracy on ImageNet at $(\epsilon=0.5,\delta=8\times10^{-7})$-DP when fine-tuning a pre-trained NFNet-F3 [2]. This result is obtained under a different assumption from the training-from-scratch setting. Specifically, it relies on **the availability of public non-private data**, or an equivalently public pre-trained model, to learn a strong representation first, and then applies DP-SGD **only during fine-tuning on the private target data** [2]. We believe this assumption is practically reasonable in some settings, and it also helps explain why studying the sub-1 regime can be particularly meaningful in that scope.
> > >
> > > In addition, DeepMind's paper [2] follows a different optimization route from adaptive clipping, which relies on a carefully tuned Wide-ResNet (WRN) recipe with large batches and several optimization improvements. Since DeepMind paper uses a **fixed clipping** threshold, SlaClip can be incorporated into that method seamlessly. We have reported a preliminary evaluation in the $\epsilon<1$ regime on CIFAR-10 in our first rebuttal to **Reviewer k7YH, Section W2,Q3**. This also further highlights that SlaClip is a plug-and-play adaptive clipping mechanism compatible with other optimization strategies.
> > >
> > > **Dicussion**
> > >
> > > We strongly agree with the reviewer’s broader point that *DP-SGD showed a fantastic step forward asking for more future work*. In this regard, the CCS 2024 paper [3] presents an interesting result. It focuses on the privacy of the **most vulnerable samples** and shows that prior empirical evaluations can substantially underestimate privacy leakage [3]. It also finds that properly tuned DP-SGD **outperforms** several SOTA heuristic privacy defenses against SOTA membership inference attacks [3]. We believe these findings further motivate continued work on principled DP-based training methods.
> > >
> > > Accordingly, we have re-evaluated the wording around "strong privacy guarantees" in our manuscript, and highlighted the importance of the ε<1 regime explicitly. We also added both the above ε<1 experiments and the discussions to better position our paper. Thank you again for helping us improve the paper.
> > >
> > > **References**
> > >
> > > [1] N. Ponomareva et al., *How to DP-fy ML: A Practical Guide to Machine Learning with Differential Privacy*, JAIR 2023.
> > >
> > > [2] S. De et al., *Unlocking High-Accuracy Differentially Private Image Classification through Scale*, 2022.
> > >
> > > [3] M. Aerni, J. Zhang, F. Tramèr, *Evaluations of Machine Learning Privacy Defenses are Misleading*, CCS 2024.

---

### Official Review · Reviewer_iPfu · 2026-03-13

**Soundness:** 3
**Presentation:** 2
**Significance:** 2
**Originality:** 3
**Overall Recommendation:** 4
**Confidence:** 3

**Summary:**

This paper proposes **SlaClip**, an adaptive clipping method for DP-SGD that aims to extract more information from the same Gaussian release used for model updates. The key idea is to encode the clipping slack, i.e. roughly $(C_t-\|g_{t,i}\|)_+$, into $K$ extra coordinates of each per-sample gradient in a way that preserves the original per-sample $\ell_2$ bound. The resulting noisy “Slack Indicator” is interpreted as a binned estimate of the CDF of clipped gradient norms and is then used to update the clipping threshold over training. Empirically, the paper compares SlaClip to several baselines on five datasets, and reports mostly competitive or improved accuracy under the matched DP-SGD setting considered in the paper.

**Compliance With Llm Reviewing Policy:**

Affirmed.

**Final Justification:**

After considering both the paper and the rebuttal, I am changing my score to **4: Weak accept**.

I find the core idea interesting, technically plausible, and sufficiently original: using slack information from the same DP-SGD release to adapt the clipping threshold is a neat methodological contribution, and adaptive clipping remains an important practical problem. My initial main concern was the empirical evaluation, especially the fairness of the comparison and the lack of a clearly careful tuning protocol across methods. The rebuttal addressed this concern to a meaningful extent by adding a smaller shared-protocol study and clarifying several presentation points. This made the empirical case more convincing and gave a better sense of the practical behavior of the method.

That said, I still view the theoretical contribution as more incremental than foundational, so for me the case for acceptance depends primarily on the empirical support. I therefore hope, and trust, that the authors will incorporate the feedback from the discussion and strengthen the final version with a more careful empirical evaluation of the method and its limitations, especially through more fine-grained hyperparameter tuning and a cleaner batch-size study relative to adaptive clipping baselines.

I would also encourage the authors to include a more detailed discussion and comparison to the commonly used “normalized clipping” approach from De et al. (2022), as this would help position the method more clearly with respect to practical DP training recipes.

Overall, the rebuttal improved my assessment enough that I can support acceptance of the paper, provided that the final version incorporates these clarifications and strengthens the empirical discussion accordingly.

**Key Questions For Authors:**

1. According to Appendix A, all methods are run with the same fixed hyperparameters and there is no apparent per-method tuning. How were the selected hyperparameters (learning rate, momentum, batch size, weight decay, scheduler settings) chosen? Unless there is a strong justification, this may invalidate the empirical conclusions, since different methods can favor different configurations.

2. Can the authors clarify the exact interpretation of $\hat s_{t,1}$ and how Eq. 13 relates to the Appendix/Algorithm 1, especially the role of $\beta$? Also, there seems to be a typo/inconsistency here that should be corrected.

3. How were the reported $\epsilon$ values chosen, especially the relatively large values for some datasets such as CIFAR-10 and in the batch-size ablation? Also, could the authors provide a non-private and/or unclipped performance reference?

4. Do I understand correctly that larger batch size requires larger $K$, which may in turn degrade the performance of SlaClip? If so, this seems like a potentially important limitation, since good DP training often benefits from larger batch sizes. Could the authors provide a cleaner large-batch study with learning-rate retuning or a justified scaling rule?

**Limitations:**

No, I would encourage the authors to include a more substantive discussion of:
- sensitivity to the choice of $C_0$,
- the weak-signal regime under tight privacy budgets or early training,
- the trade-off between larger $K$ and noisier control signals,
- possible limitations in large-batch settings.

**Strengths And Weaknesses:**

### Strengths

- The core idea is interesting and technically plausible. The paper uses the fact that Gaussian-mechanism accounting depends on $\ell_2$-sensitivity rather than output dimension, and then designs a slack encoding that preserves the same per-sample norm bound. At a high level, this is conceptually elegant.
- The method is reasonably original. The main novelty is not a fundamentally new privacy theorem, but rather a creative use of known DP machinery to encode slack information and recover a noisy CDF-like signal for adaptive clipping.
- The problem is relevant. Adaptive clipping remains an important practical issue in DP-SGD, and richer information about gradient-norm distributions could be useful in applications where a single clipped-fraction statistic is too crude.
- The paper includes several ablations rather than only a single main comparison table, which is appreciated.

### Weaknesses

**Soundness.**
I find the core idea promising, but I have several concerns about the technical presentation and, especially, the empirical evaluation.

First, the paper moves a bit too quickly between “same privacy-cost upper bound” and stronger wording that reads more like “same privacy cost” or even “identical Gaussian mechanism.” I think the upper-bound interpretation is the correct one, and the stronger phrasing should be used more carefully.

Second, Section 2 presents Step I with a fixed minibatch size $|B_t| \equiv B$. Strictly speaking, this diverges from a standard privacy moment accounting approach such as Abadi et al., where the effective batch sampling mechanism can vary across iterations. The exposition should be aligned more carefully with the actual accountant and sampling scheme used in the experiments.

Third, some parts of the controller are more heuristic than the paper sometimes suggests. In particular, $\hat s_{t,1}$ is not exactly the unclipped fraction, but rather a near-threshold interval/CDF surrogate. Relatedly, Eq. 13 appears inconsistent with the Appendix/Algorithm 1 parameterization, and the role of $\beta$ is not cleanly reconciled in the main text.

My main concern is the empirical evaluation. According to Appendix A, for most datasets all methods are run with the same optimizer configuration (same learning rate, momentum, weight decay, and cosine schedule), and there does not appear to be meaningful per-method tuning. I understand the motivation: this defines a matched regime where methods differ only in the clipping strategy. However, unless the claim is explicitly restricted to “works well under one shared optimizer recipe,” this is not enough to establish practical superiority. Different adaptive clipping methods can favor different hyperparameter settings, and without tuning, the comparison is difficult to interpret. In fact, the discussion itself partially attributes weak baseline performance to hyperparameter mismatch, which further highlights the issue.

I also found the batch-size ablation difficult to interpret. Larger batch size changes several things simultaneously: the sampling rate, achieved privacy levels, number of steps until budget exhaustion, admissible values of $K$, and the optimization regime. At minimum, the step size should be adjusted or justified, since larger batch size often allows a larger learning rate. As written, it is hard to disentangle whether the observed trends come from the SlaClip mechanism itself or from confounded optimization effects.

Finally, I would have liked a non-private and/or unclipped reference for context. This would help assess how much utility gap remains and how meaningful the gains are in practice.

**Presentation.**
The writing is mostly clear, and the overall narrative is reasonable. However, the presentation can be improved.

The notation is heavier than necessary: superscript indexing such as $g_{t,i}^{(d+K)}$, $g_{t,i}^{(d)}$, etc., makes the method harder to parse than needed. I would strongly encourage the authors to simplify this notation.

Some of the ideas were also not fully clear to me on the first read, especially the interpretation of the Slack Indicator and how it should be read from Figures 1 and 2. These visualizations could be made more intuitive.

Appendix A should provide additional implementation details, especially for the cosine learning-rate schedule. For example, what is the final learning rate? Is it 0?

There are also a few typos/inconsistencies that should be fixed:
- There seems to be a typo/inconsistency around Eq. 13 in the main text versus the Appendix/algorithm.
- Figure 1 caption formatting could be cleaned up.
- There are several minor wording/formatting issues throughout the paper; they are not severe, but a careful proofreading pass would help.

**Significance.**
I find the idea interesting and the topic worth studying. However, the significance of the claimed uniform improvement over prior work is not fully convincing to me yet.

The claim that Adap-Clip introduces privacy-loss “overhead” is true, but the original method was arguing that this overhead is often basically negligible because it privately estimates a 1D quantity. In contrast, this paper estimates a whole CDF-like object through $K$ bins, which can indeed be more informative, but may also be more costly in practice because the signal-to-noise ratio degrades as $K$ grows.

So, to me, the paper currently demonstrates a promising alternative design point rather than a clearly superior replacement for prior adaptive clipping methods. That said, I do think the method could be useful in applications where more detailed knowledge of the gradient-norm distribution is important for good performance.

**Originality.**
I think the paper is sufficiently original. The contribution is not a major new DP theorem, but rather a clever and novel way of using the same noisy release to recover additional structure about the gradient-norm distribution. This is a meaningful methodological idea, even if the underlying privacy ingredients are standard.

---

> ### Author Rebuttal · Authors · 2026-03-31
>
> Thank you for the careful and constructive review.
>
> **Clarification on the empirical evaluation**
>
> **Q1, Q3** We appreciate the reviewer's understanding that our comparison defines *a matched regime where methods differ only in the clipping strategy*, because in practical settings repeatedly tuning hyperparameters on a private dataset to obtain the best configuration may not be feasible. Despite this initial motivation, we also provide an additional empirical analysis below, as the reviewer requested. The results show that SlaClip can outperform these baselines **even under their favored settings**.
>
> For the AutoClip baseline, we evaluate SlaClip under **AutoClip’s favored settings** (same learning rate, momentum, etc.) from the original paper (Bu et al., 2023). In the table below, the **Auto-Clip**(called **AUTO-S** in the original paper), **Vanilla-Clip**, and **non-DP** columns are taken directly from Table 1 of that paper. Under this comparison, SlaClip outperforms AUTO-S on MNIST and F-MNIST, and remains comparable on CIFAR-10. The **non-DP** column provides the *non-private* reference requested by the reviewer (Q3).
>
> | Task | Model | ε | SlaClip | AUTO-S | Vanilla-Clip | non-DP (ε = ∞) |
> |---|---|---|---:|---:|---:|---:|
> | MNIST | 4-layer CNN | 3 | 98.18 ± 0.03 | 98.15 ± 0.07 | 98.04 ± 0.09 | 99.11 ± 0.07 |
> | F-MNIST | 4-layer CNN | 3 | 86.52 ± 0.15 | 86.36 ± 0.18 | 86.04 ± 0.26 | 89.57 ± 0.13 |
> | CIFAR10 pretrained | SimCLRv2 | 2 | 92.65 ± 0.16 | 92.70 ± 0.02 | 92.44 ± 0.13 | 94.42 ± 0.01 |
>
> We also use the reported DC-SGD-E and Adap-Clip results from Table IV of *Wei et al. (2025)*. Since F-MNIST is not included in that paper, we reproduced those baselines by grid search around the recommended settings. For CIFAR-10, rather than using the original ResNet setting from their paper, which yields relatively low performance, we conducted a new unified comparison under a stronger model setting (AvgPool-GAP CNN).
>
> | Task | Model | ε | SlaClip | DC-SGD-E | Adap-Clip | Vanilla-Clip | non-DP (ε = ∞) |
> |---|---|---|---:|---:|---:|---:|---:|
> | MNIST | LeNet-style CNN | 2 | 96.84 ± 0.25 | 94.19 ± 0.87 | 95.45 ± 0.29 | 95.86 ± 0.08 | 98.99 ± 0.05 |
> | F-MNIST | LeNet-style CNN | 3 | 86.88 ± 0.09 | 86.30 ± 0.48 | 81.25 ± 0.08 | 84.68 ± 0.49 | 91.99 ± 0.08 |
> | CIFAR-10 | AvgPool-GAP CNN | 9 | 68.76 ± 0.71 | 64.38 ± 0.58 | 67.65 ± 0.64 | 42.41 ± 0.12 | 77.48 ± 0.14 |
>
> We have updated Table 1 in our manuscript.
>
> **Batchsize (Q3, Q4)** Our Table 2 presents a joint study of batch size and admissible $K$, rather than a strict $K$ ablation. Its purpose is to compare SlaClip with baselines across different batch sizes. The strict $K$ ablation is **Table 4**, where batch size and privacy budget are fixed and only $K$ is varied. Table 4 validates the effectiveness of the upper bound on $K$ defined in Eq. 14, which keeps the signal-to-noise ratio within an effective range.
>
> **Does the larger admissible $K$ required by a larger batch size degrade performance?** Our conclusion is it doesn't. Even without SlaClip, increasing the batchsize does **not** always improve performance in DP-SGD. This is also consistent with *TAN Without a Burn: Scaling Laws of DP-SGD* (ICML 2023), which shows that larger batch sizes are not inherently beneficial: any apparent mega-batch advantage depends on the associated noise regime and may disappear or even reverse depending on the task. For this reason, the reviewer's statement that *good DP training often benefits from larger batch sizes* does not necessarily apply in DP-SGD.
>
> We performed a cleaner large-batch study on CIFAR-10 using a TAN-inspired scaling setup: the DP-SGD parameters were re-scaled across batch sizes, and the learning rate followed a simple linear-scaling rule, $\mathrm{LR}(B)=0.05 \cdot B/256$. Due to space limits, we place the full parameter table and the full results for $ε\in \{3,5,7\}$ in the [anonymous supplementary tables](https://github.com/SlaClip/slaclip-rebuttal-tables/blob/main/tables.md). A representative slice at ε=7 is shown below.
>
> | Method | 256 | 512 | 1024 | 2048 | 4096 |
> | --- | ---: | ---: | ---: | ---: | ---: |
> | Vanilla DP-SGD | 48.70 ± 0.22 | 60.97 ± 0.34 | 65.65 ± 0.17 | 66.63 ± 0.21 | 66.75 ± 0.18 |
> | SlaClip | 47.95 ± 0.17 | 63.33 ± 0.29 | 67.42 ± 0.80 | 69.05 ± 0.55 | 68.15 ± 0.72 |
> | non-DP | 74.90 ± 0.24 | 75.15 ± 0.26 | 76.14 ± 0.33 | 76.90 ± 0.54 | 75.23 ± 0.78 |
>
> Under this cleaner scaling setup, SlaClip is competitive at small batch size and better than vanilla DP-SGD from batch size 512 onward, while the overall trend across batch size is shared by both DP methods. This suggests that the batch size effect should not be attributed solely to the larger admissible $K$ in SlaClip.
>
> **Q2. Presentatoin and notation.** $\beta$ is the target clipping ratio thus should default to $0.5$, and $\hat{s}_{t,1}$ should be described as an estimate rather than an exact unclipped fraction.
>
> We have fixed in our manuscript all the presentation issues reported.

---

> > ### Author Rebuttal · Reviewer_iPfu · 2026-04-04
> >
> > Thanks the authors. Many of my concerns have been addressed.
> >
> > ---
> > ## Follow up for "Reply Rebuttal Comment by Authors"
> >
> > ---
> >
> > Thank you for the responses, clarifications, and additional empirical results. I appreciate the effort the authors put into strengthening the paper during rebuttal.
> >
> > However, I think my main concern is still not fully resolved, namely the issue of **careful hyperparameter tuning and fairness of the empirical comparison**. In particular, I would find it more convincing to see a **common and fair tuning protocol across methods**, even on a smaller number of representative problems (e.g. 1–3 tasks) if compute budget is a constraint. To me, this would be a stronger empirical justification than reusing hyperparameters or reported numbers from prior work.
> >
> > Relatedly, I would also value a more thorough comparison to the closest adaptive clipping baselines under such a protocol. At this stage, what matters more to me is not only whether the proposed method improves average performance across many tasks, but also whether the experiments clearly reveal the **limits of the method** and help the reader understand its practical strengths and weaknesses.
> >
> > I find the core idea interesting and worthwhile. At the same time, since the theoretical contribution appears more incremental than foundational, I think acceptance requires a particularly careful and rigorous empirical evaluation. If the fairness/tuning issue were resolved more convincingly, together with a stronger comparison to adaptive clipping baselines, I would be open to increasing my score and supporting acceptance.

---

> > > ### Author Response · Authors · 2026-04-06
> > >
> > > ### Edited after the reviewer's new follow-up comment on 06 April
> > >
> > > ---
> > >
> > > Thank you for the thoughtful follow-up. The new evaluation results are provided in [anonymous supplementary figures](https://github.com/SlaClip/slaclip-rebuttal-tables/blob/main/figures.md). A more detailed explanation is given below.
> > >
> > > **1. Fairness of our previous experiments**
> > >
> > > We would like to re-highlight that our earlier comparisons are fair in the following sense: for each task, all methods shared the same task and split, model, optimizer settings, and privacy accountant; only the clipping / threshold-update rule changed.
> > >
> > > **2. Common and fair tuning protocol**
> > >
> > > We appreciate the reviewer’s point that a more careful empirical evaluation is needed. We therefore ran a smaller shared-protocol study on representative tasks, focusing on the closest adaptive clipping baselines (Adap-Clip and AutoClip). Specifically, we swept the hyperparameters over the range used in the original baseline papers:
> > > - the learning rate over $\{0.05, 0.1, 0.2, 0.5, 1\}$,
> > > - two learning-rate schedules (constant and cosine), and the
> > > - initial clipping threshold over $C_0 \in \{0.1, 1, 10\}$,
> > >
> > > while fixing the remaining training setup across methods: same models, $\epsilon = 3$, $\delta = 10^{-5}$, 30 epochs, batch size 512, SGD with momentum 0.9, weight decay 0, and the same RDP privacy accountant. The noise multiplier $\sigma$ was calibrated separately for each dataset to match the target privacy budget rather than tuned independently.
> > >
> > > **3. Preliminary results**
> > >
> > > Given the limited time remaining in the discussion window, we nevertheless tried our best to **implement the above protocol and provide a preliminary evaluation on FMNIST and CIFAR10**. Due to space limit, the results are reported in [anonymous supplementary figures](https://github.com/SlaClip/slaclip-rebuttal-tables/blob/main/figures.md).
> > >
> > > A brief summary of the practical strengths and weaknesses is given below.
> > >
> > > **4. Practical strengths and weaknesses**
> > >
> > > *Strengths*
> > > - SlaClip can achieve the **best accuracy** within the current hyperparameter tuning region (starred results in the figures, 87.1% on FMNIST and 61.5% on CIFAR10).
> > > - Under the shared tuning protocol, SlaClip remains competitive on both representative tasks and generally outperforms the closest adaptive baseline, Adap-Clip, in the studied learning-rate region on both FMNIST and CIFAR-10.
> > > - Under both the constant and cosine learning-rate schedules, SlaClip remains stable across different $C_0$ values in the low-to-moderate learning-rate regime, e.g., \(0.05\), \(0.1\) under batch size 512. The cosine schedule further enlarges the high-performance region, especially on CIFAR-10.
> > >
> > > *Weakness / limit*
> > > - We observe that under large learning rates (e.g., lr = 1), adaptive clipping can enter an unstable regime. As shown in Fig. S1-right, accuracy drops below 80% on FMNIST under the constant learning-rate schedule. **Importantly, this behavior is not specific to SlaClip: we observe the same pattern for other adaptive clipping baselines, which all perform substantially worse than fixed clipping (Vanilla-Clip) under the same setting**. One possible explanation is that the optimization dynamics change too abruptly in early training, causing per-sample gradient norms to grow rapidly and the clipping threshold to increase sharply within the first epoch. Once this happens, subsequent threshold updates become less effective. We therefore view this as **a shared limitation of adaptive clipping** under overly aggressive learning rates, rather than a limitation unique to SlaClip. A possible practical remedy is to cap the clipping threshold when such large learning rates are used, although our current evidence suggests that this large-learning-rate regime already lies outside the range where adaptive clipping is most effective.
> > >
> > > We are continuing to incorporate a more complete version of this empirical study across more tasks, and update the manuscript so that the practical strengths and limitations of the current SlaClip clipping strategy are stated more explicitly.
> > >
> > > **5. Compatibility and significance**
> > >
> > > We would like to emphasize that SlaClip should not be viewed as a merely incremental contribution. SlaClip is foundational in this sense: it turns adaptive clipping into a reusable module rather than a task-specific redesign of DP-SGD. It makes adaptive clipping a **plug-and-play mechanism** with **no additional privacy cost**, while leaving vanilla DP-SGD unchanged. To our knowledge, **no prior adaptive clipping method** provides this same combination. As a result, DP-SGD-based methods that do not already use adaptive clipping can adopt SlaClip directly. Our first response to Reviewer k7YH (W2, Q3) provides a preliminary compatibility study illustrating this point.
> > >
> > > We hope this fully addresses the reviewer's concern.

---

### Official Review · Reviewer_k7YH · 2026-03-22

**Soundness:** 3
**Presentation:** 3
**Significance:** 2
**Originality:** 1
**Overall Recommendation:** 3
**Confidence:** 5

**Summary:**

This paper revisits the hyper-parameter tuning in DP-SGD. The authors address the problem of selecting and updating the gradient clipping threshold, which is traditionally fixed and can severely degrade model utility if misaligned with the gradient norm distribution. The SlaClip proposed leverages slack, the difference between an unclipped gradient's norm and the clipping threshold, by encoding this information into extra gradient dimensions. This enables a same-query release of a "Slack Indicator" that estimates the cumulative distribution function (CDF) of clipped gradient norms, guiding the threshold adaptation without expending additional privacy budget.

**Compliance With Llm Reviewing Policy:**

Affirmed.

**Key Questions For Authors:**

1. Based on the gradient histogram, I think a deeper exploration on how to select the clipping threshold can be done. There has been a long line of works studying the bias incurred by clipping, for example, "Understanding gradient clipping in private sgd: A geometric perspective", "A theory to instruct differentially-private learning via clipping bias reduction". A more reasonable clipping strategy is to minimize the clipping bias based on the histogram data.

2. Is that necessary to perform the histogram estimation in every single iteration? Would there be more sharpened utility-privacy to include the histogram every 5 or 10 iterations?

3. Can you show improvement over the SOTA DP-SGD performance?

**Limitations:**

please see above

**Strengths And Weaknesses:**

Strengths:
+ The privately-released gradient histogram seems interesting.

Weaknesses:
+ Technically, there is no essential difference between the concatenation of the one-hot histogram representation and the classic composition applied in the prior work for the hyper-parameter. I think the contribution about the histogram estimation is a bit overstated. The release of additional dimension does not come for free, though I think it does have some advantage in a sense of automatic normalization accompanying with the per-sample gradient.
+ It is not clear whether the other hyper parameters, for example learning rate, have been optimized. The baseline, for example, on CIFAR10, is much worse than the SOTA work, for example, "Unlocking high-accuracy differentially private image classification through scale" where for $\epsilon = 8$, the test accuracy can be above 80%.

---

> ### Author Rebuttal · Authors · 2026-03-31
>
> Thank you for the careful review.
>
> **Clarification of novelty (W1).** SlaClip is a **novel plug-and-play adaptive-clipping mechanism**, rather than a traditional histogram-based method. It has two core advantages over existing techniques: (1) the Slack Indicator **does not spend additional privacy budget** when estimating the histogram of gradient norms (as shown by Theorem 3.1 and Lemma 3.2). In contrast, the traditional methods such as Adap-Clip and DC-SGD suffer from what we believe is a *key weakness*: they necessarily incur extra privacy cost due to additional queries, such as the bit-sum count of clipped samples. SlaClip is proposed as *an elegant approach to cleanly overcome that weakness*, as noted by Reviewer ATRi (Strength 1). (2) SlaClip is compatible with vanilla DP-SGD and its variants because the standard DP-SGD average-and-noise query remains unchanged. This means that SlaClip can be plugged into other DP-SGD optimization mechanisms that currently use fixed clipping, e.g., the DeepMind 2022 paper *Unlocking High-Accuracy …* mentioned in the review; see Section **W2, Q3** below for an empirical analysis. In summary, traditional adaptive clipping methods either incur additional privacy cost or are not directly compatible with such methods.
>
> It follows that if the reviewer's statement *"the release of additional dimension does not come for free"* refers to **privacy cost**, then it is a misunderstanding. If instead the concern is about the **computational cost**, we maintain that the overhead is modest (e.g., about 10% additional time per epoch relative to vanilla DP-SGD), as detailed in our response to Reviewer uJLz (Section **CIFAR-10 Runtime Comparison**).
>
> **W2, Q3.** Thank you for pointing us to this DeepMind 2022 paper. We did not include it as a baseline because it follows a **different optimization route** from ours. SlaClip focuses on adaptive clipping, whereas DeepMind 2022 reports 81.4% on CIFAR-10 using a carefully tuned Wide-ResNet (WRN) recipe with large batches and several optimization improvements. Since DeepMind 2022 uses fixed clipping, we rather compare its original recipe with **the same recipe augmented by SlaClip to enable adaptive clipping**. We present a preliminary comparison below.
>
> | ε | DeepMind 2022 baseline | Same recipe + SlaClip |
> |---|-----------------------------:|----------------------:|
> | 0.2 | 12.91 | 17.66 |
> | 0.3 | 20.40 | 27.16 |
> | 0.5 | 25.43 | 29.80 |
>
> Due to the design of DeepMind 2022, the computational cost is substantial (e.g., over 730 seconds per batch on our A100 GPU, requiring 400+ hours to reach ε=8). As a result, the full experiment was still running at the rebuttal deadline. Even so, the preliminary results already show consistent performance gains from SlaClip.
>
> **Q1.** Clipping-bias minimization can be a theoretically appealing alternative to the dynamic target-ratio controller used by SlaClip. While developing the threshold update rule, we also implemented and evaluated such a bias-minimization target, following the ideas in *A theory to instruct...*. However, since this in practice did not perform well enough relative to our current method, those preliminary runs were not included in the submission. A compact comparison is shown below.
>
> | Threshold update rule | MNIST | CIFAR-10 |
> | --------------------- | ----- | -------- |
> | Bias-minimization target (preliminary) | 93.58 ± 0.35 | 67.43 ± 0.62 |
> | SlaClip | 96.84 ± 0.22 | 68.76 ± 0.71 |
>
> One possible explanation is that the optimal clipping target itself changes over the course of the training, while the clipping bias must be inferred indirectly from noisy histogram bins rather than observed directly. This likely makes the update more sensitive and less stable, especially early in the training, when the gradient-norm distribution changes rapidly.
>
> **Q2.** Thank you for this helpful suggestion regarding less frequent slack updates. As clarified above, changing the update frequency does not affect the additional privacy cost, which remains zero. We conducted the requested ablation to evaluate its practical impact on utility. The results are shown below.
>
> | Dataset | ε | Every iter. | Every 5 iters. | Every 10 iters. |
> |---|---:|---:|---:|---:|
> | **MNIST** |  |  |  |  |
> |  | 1 | 64.37 ± 4.54 | 63.85 ± 2.43 | 62.61 ± 3.70 |
> |  | 2 | 96.21 ± 0.35 | 95.32 ± 0.63 | 95.26 ± 0.57 |
> |  | 3 | 96.84 ± 0.25 | 96.29 ± 0.25 | 96.26 ± 0.20 |
> | **CIFAR-10** |  |  |  |  |
> |  | 5 | 59.25 ± 1.29 | 57.59 ± 0.49 | 57.18 ± 1.18 |
> |  | 7 | 65.35 ± 1.09 | 63.20 ± 0.53 | 63.23 ± 0.52 |
> |  | 9 | 68.76 ± 0.71 | 67.79 ± 0.54 | 67.55 ± 0.15 |
>
> These results suggest that our per-iteration histogram estimation is beneficial, as it consistently performs best. A likely reason is that updating only every 5 or 10 iterations cannot track the changing gradient-norm distribution as accurately, especially when the distribution evolves quickly during training.

---

> > ### Author Rebuttal · Reviewer_k7YH · 2026-04-03
> >
> > I thank the authors for their detailed response.
> >
> > The additional experiments successfully demonstrate the compatibility of slaclip with other DP-SGD techniques. However, I maintain several technical concerns regarding the slaclip algorithm itself.
> >
> > First, to clarify my previous comment: both concatenation and composition will asymptotically consume the same privacy budget. I understand the core trick in slaclip—that for per-sample gradients with an $L_2$ norm smaller than the clipping threshold, the remaining "gap" can be privately released without further utility loss to the gradient estimation. However, it is unclear how this translates to clipping bias reduction via adaptive threshold selection, since we have no information regarding the gap for gradients whose norms exceed the threshold.For example, suppose 99% of the per-sample gradients have an $L_2$ norm above 5, but the current clipping threshold is 1. Even with perfect knowledge of the gradient norm histogram below 1, the paper does not explain how this limited information can meaningfully guide the selection of the next threshold. Most existing adaptive clipping methods estimate statistics for the entire batch (e.g., the mean or median), rather than only the subset of small gradients. If slaclip ultimately requires estimating statistics for the full batch, the advantage of the proposed concatenation method over standard composition is not immediately clear. This is why I suggested providing a more comprehensive study detailing the relationship between clipping bias and the proposed adaptive clipping mechanism.
> >
> > Second, the mechanics of the slaclip algorithm in Equations (6)–(8) are difficult to follow, particularly the introduction of the all-ones vector. If the goal is to estimate the histogram, why not simply concatenate a normalized one-hot vector to the per-sample gradient? I suspect there is a specific technical rationale for this, but the current presentation obscures it. Providing a concrete, step-by-step example would significantly improve the clarity of this section.

---

> > > ### Author Response · Authors · 2026-04-06
> > >
> > > Thank you for the follow-up. We are glad that the additional experiments successfully addressed your concerns. Below we clarify the remaining technical points.
> > >
> > > ---
> > >
> > > **1. Difference between SlaClip and traditional methods**
> > > > Reviewer's comment "First...both concatenation and composition will asymptotically consume the same privacy budget."
> > >
> > > We respectfully **disagree** with this characterization in our setting. SlaClip does **not** use a naive concatenation of auxiliary statistics to the per-sample gradient. It is designed with **two key properties** (Section 3 Overview, Line 151):
> > > - **Property 1** each extended per-sample gradient still has $\ell_2$ norm at most $C$;
> > > - **Property 2** the auxiliary information is embedded in the **same query**, not released via an additional private query.
> > >
> > > Hence SlaClip preserves the original global $\ell_2$ sensitivity and thus the same per-step privacy-cost upper bound as vanilla DP-SGD (Theorem 3.1, Lemma 3.2).
> > >
> > > **So, as agreed by other reviewers, enabling SlaClip adds no privacy cost beyond vanilla DP-SGD.**
> > >
> > > We provide a [step-by-step toy example](https://github.com/SlaClip/slaclip-rebuttal-tables/blob/main/tables.md) in the Appendix to clarify this construction.
> > >
> > > > Reviewer's comment "..., the remaining "gap" can be privately released without further utility loss to the gradient estimation."
> > >
> > > As noted above, a more accurate statement is **"without incurring additional privacy cost,"** not "utility loss." Traditional composition-based methods such as Adap-Clip do not satisfy Property 1 or 2, and therefore incur an additional privacy cost $ε_e$. The relevant per-step privacy cost is therefore:
> > >
> > > - **Vanilla DP-SGD:** $ε_v$
> > > - **SlaClip:** $ε_v$
> > > - **Adap-Clip:** $ε_v + ε_e$
> > >
> > > Under a fixed total privacy budget equal to $ε_v$, Adap-Clip must therefore **calibrate a larger noise scale for the averaged gradients** to absorb the extra cost $ε_e$, whereas SlaClip does **not** need this step.
> > >
> > > ---
> > >
> > > **2. How SlaClip updates $C_t$**
> > >
> > > > Reviewer's comment “If slaclip ultimately requires estimating statistics for the full batch... is not immediately clear.”
> > >
> > > SlaClip does **not** require full-batch statistics. More importantly, Slack Indicator estimates the **CDF of $clip(||g_{t,i}||)$, not the histogram of gradient norms below $C$**, where $clip(||g_{t,i}||)$ refers to the gradient norms **after** current clipping, as illustrated in Fig. 1.
> > >
> > > As explained in line 191, the first coordinate of the Slack Indicator, $\hat s_{t,1}$, already estimates **the fraction of unclipped samples**, while other coordinates capture cumulative probabilities for smaller gradients. Once these cumulative probabilities are available, SlaClip can tell whether the current $C$ is too small and update it accordingly. In this sense, SlaClip does **not** estimate clipping bias directly; it uses the estimated unclipped fraction and related cumulative probabilities as the signal for threshold adaptation. See the concrete example below:
> > >
> > > > Reviewer's comment "For example, suppose 99% of the per-sample gradients ... guide the selection of the next threshold."
> > >
> > > This is exactly the noise-dominated regime in Fig. 2-A. If 99\% of per-sample gradients satisfy $\||g_{t,i}\|| > 5$ while $C_t = 1$, then almost all slacks are zero, so the estimated CDF of uclipped gradients is nearly zero everywhere and the Slack Indicator is noise-dominated.
> > >
> > > This still gives a clear update signal. It means that almost no samples are currently unclipped, i.e., the current $C_t$ is far too small and clipping is too aggressive. In this regime, SlaClip follows the geometric update rule (Eq. 13) to increase $C_t$ (in our Fig.2 example, from 1 to 1.284), and continues doing so until enough samples fall near or below the threshold. Only then does the estimated CDF become informative for finer adaptation.
> > >
> > > ---
> > >
> > > **3. Why not concatenate a normalized one-hot vector**
> > >
> > > > Reviewer's comment “If the goal is to estimate the histogram, why not simply concatenate a normalized one-hot vector to the per-sample gradient?”
> > >
> > > A normalized one-hot concatenation would directly **violate Property 1**. For example, concatenating a one-dimensional gradient $[0.9]$ with a one-hot vector $[1]$ increases the $\ell_2$ norm from $0.9$ to $\sqrt{0.9^2+1^2}\approx 1.34$. This breaks the privacy-cost upper bound of vanilla DP-SGD and therefore violates SlaClip's design goal.
> > >
> > > **4. Readability of Equations**
> > >
> > > > Reviewer's comment “Second, the mechanics of the slaclip algorithm in Equations (6)–(8) ... clarity of this section”
> > >
> > > Eq.(6)-(8) are designed to ensure that **the Slack Indicator satisfies Lemma 3.2**. The all-ones vector makes each sample contribute to all bins below its slack level. Concretely, this construction is what allows the Slack Indicator to satisfy Properties 1 and 2.
> > >
> > > Following the reviewer's suggestion, we have added the mentioned concrete example to clarify this construction.
> > >
> > > We hope these points fully address the remaining concerns.

---

### Decision · Program_Chairs · 2026-04-30

**Decision:**

Accept (spotlight)

**Comment:**

The paper proposes a new algorithm for adaptive clipping in DP-SGD, called SlaClip. The key idea is to append extra coordinates to each clipped gradient that encode the “clipping slack.” This makes it possible to estimate the percentile norm within a batch and adapt the clipping threshold accordingly. Importantly, the $l_2$-norm of the augmented gradient remains unchanged, so the execution of DP-SGD is unaffected while providing this information essentially for free, not causing any additional privacy loss. Empirically, the paper evaluates the method on five datasets and shows improved accuracy as well as robustness to several hyperparameter choices, such as the clipping threshold and batch size.

Overall, the paper provides an elegant and promising approach to adaptive clipping in DP-SGD and is of interest to machine learning community.

In their final version, authors should address all concerns raised by the reviewers, such as adding the experimental results with fairly tuned hyperparameters, and add discussion for the small epsilon<1 case.